# Caseins: Versatility of Their Micellar Organization in Relation to the Functional and Nutritional Properties of Milk

**DOI:** 10.3390/molecules28052023

**Published:** 2023-02-21

**Authors:** Ashish Runthala, Mustapha Mbye, Mutamed Ayyash, Yajun Xu, Afaf Kamal-Eldin

**Affiliations:** 1Department of Biotechnology, Koneru Lakshmaiah Education Foundation, Vijayawada 522302, India; 2Department of Food Science, United Arab Emirates University, Al Ain P.O. Box 15551, United Arab Emirates; 3Department of Nutrition and Food Hygiene, School of Public Health, Peking University, Beijing 100871, China; 4Zayed Bin Sultan Center for Health Sciences, United Arab Emirates University, Al Ain P.O. Box 15551, United Arab Emirates

**Keywords:** milk proteins, casein, casein micelles, cow, camel, human, African elephant, structures, functional properties, nutritional properties

## Abstract

The milk of mammals is a complex fluid mixture of various proteins, minerals, lipids, and other micronutrients that play a critical role in providing nutrition and immunity to newborns. Casein proteins together with calcium phosphate form large colloidal particles, called casein micelles. Caseins and their micelles have received great scientific interest, but their versatility and role in the functional and nutritional properties of milk from different animal species are not fully understood. Caseins belong to a class of proteins that exhibit open and flexible conformations. Here, we discuss the key features that maintain the structures of the protein sequences in four selected animal species: cow, camel, human, and African elephant. The primary sequences of these proteins and their posttranslational modifications (phosphorylation and glycosylation) that determine their secondary structures have distinctively evolved in these different animal species, leading to differences in their structural, functional, and nutritional properties. The variability in the structures of milk caseins influence the properties of their dairy products, such as cheese and yogurt, as well as their digestibility and allergic properties. Such differences are beneficial to the development of different functionally improved casein molecules with variable biological and industrial utilities.

## 1. Introduction

The term casein was derived from the Latin word *caseus*, meaning cheese. Four casein protein families (κ-, β-, αS1-, and αS2-caseins) have evolved in different mammalian species to maintain specialized roles in milk [1], and their primary function is the provision of nutrients and minerals, especially calcium, to offspring while maintaining fluidity in mammary glands. In addition, caseins are the major milk proteins that provide amino acids as well as immunity to infants, but their functions are significantly affected by their digestibility [2]. In different mammalian species, caseins have been subjected to several evolutionary modifications of their primary sequences and posttranslational modifications by phosphorylation and glycosylation, inducing an overall change in their structural and functional properties. For example, phosphorylation of α- and β-caseins as well as glycosylation of κ-casein are critical modifications, affecting the formation and stability of casein micelles [3].

Caseins are associated with each other and with calcium phosphate in the form of colloidal nanostructures called micelles. Together with the fat globules, whey protein, and minerals in milk, these micelles play important roles in preventing the precipitation of casein proteins and in stabilizing milk as an emulsion [4]. Knowledge about casein micelle formation is limited mainly to bovine milk, which is not even fully understood. According to the available knowledge, αS1-, αS2-, and β-caseins are involved in calcium-binding and are mainly concentrated in the interior of spherical micelles. The calcium-sensitive caseins (αS1-, αS2-, and β-) bind calcium, mainly by electrostatic interactions with colloidal calcium phosphate (CCP), and further aggregate via weaker interactions, including hydrophobic and van der Waals interactions, as well as hydrogen and ionic bonding [5,6]. In addition, κ-casein forms a type of “brush” on the surface of the micelle that interacts with the whey and ensures electrostatic repulsion between micelles [7].

As shown in Table 1**,** human and African elephant milks are the most similar among these milks, both of which lack αS1- and/or αS2-caseins, indicating that not all mammals require the presence of all four caseins [8,9]. Goat, sheep, buffalo, camel, and horse milks are closely related to bovine milk. However African elephant’s milk is uniquely different: it contains 82.44% water, with 17.56% total solids containing 5.23% protein, 15.10% fat, and is rich in lactose and oligosaccharides [10]. Compared to the other species’ milks, African elephant milk contains higher levels of glucosamine, which is important for bone growth because amino sugars play significant roles in chondrocyte production [11]. It’s the only mammal in the study where both α-CNs are absent in the casein profile. The organization of caseins in micelles is governed by their hydrophobic, electrostatic, and steric properties, leading to a delicate balance between strong and flexible interactions [12]. Different theories have been developed to explain the organization of caseins inside the bovine casein micelle, including coat-core models, submicelle models, and internal structure models [13]. These models are all based on data related to bovine milk casein micelles without the consideration of micelles of other milks, of which the information is limited. The casein structure, composition, and organization in micelles have important implications in milk coagulation in the stomach and protein digestibility, as well as in cheese making, yogurt making, and allergenicity [14].

In this review article, we compare the structures and properties of the four caseins (αS1-, αS2-, β-, and κ-caseins) present in the milks of four different mammalian species, namely cow, camel, human, and elephant. These animal species were selected for detailed comparison due to their marked differences in the presence/absence and relative ratios of the four caseins (Table 1). Key sequence variations and relative ratios of the different caseins are responsible for the different structural, biophysical, and chemical behaviors of these milks. The key physicochemical properties of the proteins are estimated to further correlate the structural disorder and percentage of secondary structures, supporting the predicted tertiary models. Combined with other prior knowledge from the literature, this approach highlights certain questions that lead to new conclusions. The discussed properties are not only responsible for their structural properties in cheese making but have a far-reaching implication in their digestibility and allergic nature. Experimental assessment and evolution of such properties will have a phenomenal role in developing functionally improved casein molecules of higher biological and industrial utility.

## 2. Genetics and Biosynthesis of Milk Caseins

### 2.1. Genetics

Caseins belong to the family of phosphoproteins synthesized in the mammary gland, and they are secreted as roughly spherical, polydisperse (50–600 nm with an average of 200 nm), supramolecular, colloidal aggregates named micelles [15,16]. Caseins are encoded by single autosomal genes, namely *CSN1S1* (αS1-casein), *CSN2* (β-casein), *CSN1S2* (αS2-casein), and *CSN3* (κ-casein), organized as a casein gene locus cluster in a DNA stretch of approximately 250 kb located on chromosome 6 [17]. In all the four selected species, the first two genes, *CSN1S1* and *CSN2*, are close to each other, *CSN3* is the most distant (especially in the elephant), and *CSN1S2* lies in between (Figure 1). The four genes are highly conserved and tightly clustered in camels, whereas they are significantly distant in elephants. Variants (or isoforms) of these genes can result from single nucleotide polymorphisms (SNPs) and nucleotide insertions or deletions. In dromedary camels, the most polymorphic gene is *CSN1S1* (248 SNPs), and the least polymorphic gene is *CSN1S2* (79 SNPs) [17]. Bovine and camel milks contain αS1-, αS2-, β-, and κ-caseins but at the different relative ratios of 38:10:40:12 and 22:9:65.5:3.5, respectively (Table 1). Human milk lacks αS2-casein, and the relative ratio of αS1-, β-, and κ-caseins in human milk is 3:70:27. African elephant milk lacks both αS1- and αS2-caseins, and it only contains β- and κ-caseins at a ratio of 89:11 [9]. The lack or low level of α-S caseins in the human and African elephant milks supports the suggestion that only κ-casein and an ancient β-casein-like protein are obligatory to form stable casein micelles, and that αS1- and αS2-caseins have developed in some species later during evolution [18]. The reason(s) behind the genetic variability of caseins and their implications on the technological and nutritional quality of milk is not yet understood and deserves further investigation [15].

### 2.2. Biosynthesis of Milk and Milk Caseins

Milk is secreted by the mammary glands of animals to provide their offspring with the macronutrients (protein, lipids, and lactose) and micronutrients (minerals and vitamins) required for growth, as well as other components needed to boost immunity, especially during the early stages of life [20]. Different animal species have different nutritional and physiological needs, making milk composition species-specific although sharing some commonalities [21]. During lactation, the mammary epithelial secretory cells secrete huge quantities of the nutrient molecules that make up milk, i.e., proteins, fat globules and soluble components, such as lactose and minerals, while others are transferred from the blood [22]. Proteins are formed in the endoplasmic reticulum by combination of the signal peptides with constituent amino acids and then transported to the Golgi apparatus for posttranslational modification by phosphorylation and glycosylation. The biosynthesis of milk proteins involves complex interactions between insulin, which controls systemic energy status, and intracellular AMP:ATP ratio, which is an indicator of the local energy status, and their regulation of mammalian target of rapamycin (mTOR) mediates the translation initiation and elongation rates of mRNA [23]. Four amino acids, namely, histidine, lysine, methionine, and leucine, are the main limiting essential amino acids involved in regulating milk protein synthesis via mTOR [24]. The gene responsible for the expression and synthesis of casein is signal transducer and activator of transcription 5 (*STAT5*) whose cooperative interaction with glucocorticoid receptor (GR) and CCAAT/enhancer-binding protein-β (C/EBPβ) drives the transcription of β-casein [25]. Differences leading to the various levels of αS1-, αS2-, β-, and κ-caseins in the milks of different animal species are not yet known.

The phosphate groups of the caseins are esterified as monoesters on serine (and to a lesser extent on threonine) with the specific sequence of Ser-X-A (where A is an anionic residue, i.e., Glu, Asp or SerP; and X is any amino acid). Most of the phosphoserine residues in caseins occur in clusters that mainly bind calcium. Glycosylation of threonine residues in proteins may include galactose, galactosamine, and/or N-acetylneuraminic (sialic) acid that occur as tri- or tetrasaccharides [26]. Several of the minor whey proteins are recruited directly from the blood to epithelial secretory cells into milk by passive diffusion or internalization [27]. The essentiality of κ-casein for casein micelle stabilization in the alveolar lumen, as well as for lactation and reproduction, has been confirmed through κ-casein gene null mutations in mice [28,29]. The current κ-casein variants of the *Bos* species may have developed from an ancestral wild type [30].

It has been suggested that casein micelles are highly disordered in non-fixed dynamic structures, in which the different caseins adopt flexible positions while maintaining an overall shape and coherence of the micelle [31]. This feature allows rapid and differential evolution of the various caseins in the different mammalian species. It has recently been shown that caseins interact and form networks with the major whey proteins (β-lactoglobulin, α-lactalbumin, lactoferrin, and serum albumin), and these networks, in turn, interact with the other proteins [32]. The stability of the casein micelles and their destabilization are important for the coagulation of milk in the stomach during digestion as well as in the processing of milk to make yogurt and cheese [32]

## 3. Molecular Structures of Milk Caseins

### 3.1. Amino Acid Sequence Identities, and Instability

Caseins are single-chain polypeptides differing in length and amino acid sequence. Milk proteins, in general, and caseins exhibit high heterogeneity due to the existence of different isoforms with variable amino acid sequences. The currently known bovine milk casein variants include 15 variants of β-casein (A1, A2, A3, B, C, D, E, F, G, H1, H2, I, J, K, and L), 11 variants of κ-casein (A, B, C, E, F1, F2, G1, G2, H, I, and J), 10 variants of αS1-casein (A, B, C, D, E, F, G, H, I, and J), and 5 variants of αS2-casein (A, B, C, D, and E) [33], some of which may be involved in posttranslational modification(s) by phosphorylation and/or glycosylation. Some genetic variants have also been identified in other animals, but the available data are not conclusive. Moreover, the genetic factors responsible for these variations are unknown.

A multiple sequence alignment of four major casein variants (αS1-, αS2-, β- and κ-) has been shown in Figure 2 for the milks of four mammals (cow, camel, human, and African elephant). The four caseins show diverse sequence similarity within the dataset, while their signal peptides share a significantly high sequence similarity (Table 2). In generally, bovine αS1-, αS2-, β-, and κ-caseins showed the lowest similarity to those of the other species. The average sequence similarity and standard deviation of the sequence similarity within the αS1-, β- and κ-casein sets were 45.9 ± 5.5, 59.9 ± 4.5, and 56.2 ± 4.0, respectively. Moreover, human β-casein was more evolutionarily conserved with an average sequence identity of 60.526 compared to the other three functional homologues. PRALINE analysis revealed a strong conservation of the casein sequences across the four evolutionarily distant organisms (Figure 2). To effectively assess the level of such functionally conserved residues, the positions with a conservation score of at least 7 of 10 were considered. The αS1-, αS2-, β- and κ-casein sequences had conservation scores of 7, 8, 9 and 10, respectively, with the following number of conserved residues in the species: αS1-casein with 17, 2, 18, and 19 conserved residues in bovine, camel, human, and African elephant milks, respectively; αS2-casein with 4, 1, 12, and 3 conserved residues in bovine, camel, human, and African elephant milks, respectively; β-casein with 1, 0, 6, and 4 conserved residues in bovine, camel, human, and African elephant milks, respectively; and κ-casein with 59, 126, 87, and 60 conserved residues in bovine, camel, human, and African elephant milks, respectively. As nature tends to pack the function into the smallest possible protein sequence length to conserve energy and resources, the bovine, camel, human, and elephant proteins, encoding 185, 193, 223, and 181 amino acids, respectively, are considered to normalize the conservation across the constructed alignments, which implies that 81 (43.8%), 129 (66.8%), 123 (55.1%), and 86 (47.5%) of the amino acids, respectively, play a crucial role in the functionally of casein molecules.

The highly conserved sequences of milk caseins include mainly the signal peptides, the calcium-sensitive SSSEE motif, the P/Q residues, and the hydrophobicity in the C-termini (Figure 2). The first 15 amino acids in αS1-, αS2-, and β-caseins and the first 21 amino acid residues in κ-casein represent the signal peptides (Table 2), which are required in the presecretory proteins for crossing the endoplasmic reticulum membrane of the secretary cell. The high homology among the signal peptides of β-, αS1-, and αS2-caseins (~70%), which have evolved from a common ancestor, is caused by the sharing of common sequences of clustered phosphoseryl and glutamyl residues [38]. At the time of secretion, signal peptidases remove the signal peptides from the rest of the polypeptide chains that will be transferred across the membrane, which are referred to as the primary sequence of the “mature” proteins in the milk. The SSSEE motif, which is vital for calcium binding through the phosphate groups attached to its seryl residues [39], is conserved in the caseins of the four studied animals (Figure 2). In addition, the proline and glutamine residues in the C-termini of these caseins support their intra- and inter-hydrophobic interactions [39,40]. Unlike globular proteins, the conservation of tryptophan, cysteine, tyrosine, and phenylalanine residues is low in caseins [41]. In addition to their calcium-binding role, αS1- and β-caseins act as chaperon proteins that prevent the aggregation of several other proteins [42]. For example, β-casein inhibits the aggregation of κ-casein, α-lactalbumin, insulin, lysozyme, alcohol dehydrogenase, and catalase, by forming stable complexes with denaturing substrate proteins [43,44]. The chaperone activity of αS1- and β-caseins is facilitated by the high frequency of proline residues, which is accompanied by low net charge and low charge density [45]. These two features, i.e., a SSSEE motif in the N-termini and hydrophobicity in the C-termini, are essential for the formation and stabilization of casein micelles.

### 3.2. Distribution of Uncharged Polar, Hydrophobic, Aromatic, Isoelectric Points and Polar Amino Acids in Caseins

A similarity trend for the uncharged polar, hydrophobic, aromatic, hydrophilic polar, and three other encoded residues across the compared proteins is shown in Figure 3. The functionally crucial residues are likely to be conserved across certain specific positions (Figure 2). As sequence similarity may be due to natively encoded signature patterns across these sequences, and because these proteins have not been experimentally resolved thus far, it is important to decipher the key segments that affect milk quality with reference to nutritional value and processing potentials. The isoelectric point (pI) of proteins is the pH value corresponding to net charge of zero over the whole protein. Because caseins are natively unfolded proteins, their theoretical pI values can be computed from the amino acid sequences and their pKa values using the Henderson–Hasselbalch equation [46]. The studied caseins differ considerably in their pIs, which vary from 4.98 for bovine αS1-casein to >8.5 for bovine αS2-casein and all κ-caseins, except bovine κ-casein. Thus, most of the caseins are moderately acidic, except αS2-casein of bovine and κ-caseins of camel, human, and African elephant. The pI values are useful for analyzing the charge stability of the molecules [47].

Protein hydropathy plots (Figure 4) are useful fingerprints that can be used to compare the structural content and spatial distribution of the different amino acids in caseins according to their hydrophobic and hydrophilic scores. The hydropathy profile, i.e., the patterns of hydrophilicity and hydrophobicity, as well as the three-dimensional structural plots, supplement the information provided by the amino acid sequence of a protein [48]. The construction of hydropathy plots for a protein sequence is based on the classification of the amino acids into three groups as follows: (i) polar: D, E, H, K, N, Q, and R; (ii) hydrophobic: A, F, I, L, and M; and (iii) weakly hydrophilic, weakly hydrophobic, or ambiguous amino acids: S, T, W, and Y. Each amino acid is given a hydropathy score between +4.6 (most hydrophobic) and −4.6 (most hydrophilic) [49]. Proline and glycine are excluded because their unique backbone properties are more important than their hydropathies, while cysteine is excluded because its oxidized form lacks the polarizable sulfur [50]. The undulations across the hydropathy profile of these proteins are highly similar, thus uncovering a few key points.

Although the scoring value may be altered for isolated amino acids, it estimates the molecular properties and unveils some key similarity features. Firstly, camel and human αS1-caseins show lower and comparatively similar scores compared to bovine milk, which has a higher-than-average score for αS2-casein. Secondly, bovine β- and κ-caseins show substantially lower hydropathy indices compared to the other organisms whose average scoring undulations fall below −0.3. Thirdly, unlike other organisms, bovine caseins show a remarkably similar lower scoring pattern. The bovine, camel, and human αS1-caseins show total hydropathy scores of −110.034, −155.352, and −141.474, respectively, compared to the scores of −163.489 and −129.012 for the bovine and camel αS2-caseins, respectively. Moreover, the bovine, camel, human, and African elephant β-caseins show total scores of −53.08, −59.216, −32.166, and −61.17, respectively, and the bovine, camel, human, and African elephant κ-caseins show total scores of −63.212, −32.135, −52.888, and −38.743, respectively. Lastly, the conserved hydrophobic region in the C-terminal of all these caseins should be functionally responsible for their common characteristic features. The surface hydrophobicities of bovine and camel β-caseins have recently been compared at different pH values (3, 6, and 9) and temperatures (25, 65, and 95 °C) [53]. Because the electrostatic charges are negative and surface hydrophobicity is pH-independent above the isoelectric point, surface hydrophobicity is higher for bovine casein than for camel casein at pH 3 but nearly similar at pH values of 6 and 9. The higher hydrophobicity of bovine β-casein can be explained by the presence of one tryptophan residue and one tyrosine residue in the hydrophobic portion of this protein compared to five tyrosine residues and eight phenylalanine residues but no tryptophan residues in camel β-casein [54,55]. Temperature has a minimal effect on β-casein hydrophobicity, and this effect is less important for camel β-casein than for bovine β-casein [56].

Hydrophobic interactions play major roles in stabilizing the native structure of proteins in an aqueous environment by packing the nonpolar side chains into the compact core of the protein to avoid contact with water. Entropy stabilization of protein structures is achieved via disulfide bonds, hydrogen bonds, hydrophobic interactions, and van der Waals interactions, which make folded proteins more stable than unfolded proteins [57].

Protein is regarded to be stable when the instability index is less than 40 [58]. Table 3 shows that κ-caseins, α-caseins, and β-caseins possess scores greater than 40, indicating instability, however, the κ-caseins had lower instability index than α-caseins, and β-caseins. Furthermore, cow milk protein was more stable than elephant, camel, and human milk proteins. Of note, α-caseins are highly diverse and exhibit a broad range of instability in the selected milks, thereby warranting further investigation. The instability of milk caseins is consistent with their unfolded structures.

### 3.3. Posttranslational Phosphorylation and Glycosylation of Milk Caseins

In the posttranslational modification of caseins, serine (Ser) residues can be phosphorylated and/or O-glycosylated, while threonine (Thr) residues can only be O-glycosylated (Figure 5). The phosphorylation of caseins affects several of their properties, including calcium binding, micelle stabilization, interactions with proteins, interactions with other molecules, and biological activities [61]. In bovine milk, the phosphorylation of caseins is highly variable with up to 3 phosphate groups on κ-casein, 5 phosphate groups on β-casein, 8 or 9 phosphate groups on αS1-casein, and 10–14 phosphate groups on αS2-casein [62]. The electrophoretic mobility of β-, αS1-, and κ-caseins is slower for camel milk than for bovine milk, suggesting a lower net negative charge [63]. Recently, various isoforms have been reported for camel milk as follows: κ-casein with no phosphate group or a single phosphate group; β-casein with 2, 3, or 4 phosphate groups; αS1-casein with 5, 6, or 7 phosphate groups; and αS2-casein with 7, 8, or 9 phosphate groups [64]. Human β-casein comprises six isoforms having 0–5 phosphate groups per molecule [56].

With preferential phosphorylation at 2–4 sites, the preferential order of phosphorylation for the human β-casein is Ser24 > Ser25 > Ser23 > Ser21 > Thr18 [66]. The African elephant β-casein has been reported to have five isoforms, and one isoform is characterized by a single phosphorylated Ser9 residue and by truncation of the ESVTQVNK peptide sequence, shortening its length to 200 amino acid residues in comparison to the 208 residues of the other four unphosphorylated forms [9]. Human αS1-casein has been described to have three phosphorylation variants—no phosphorylation, one phosphate group at Ser18, and two phosphate groups at Ser18 and Ser26—a pattern that is different from that in ruminants where extensive phosphorylation occurs in the Ser70–Glu78 conserved region [67].

Bovine κ-casein residues, including Thr154 (41%), Thr163 (29%), Thr152 (14%), Thr142 (7%), and Thr157 (0.1%) [68], can be O-glycosylated with the following molecules: disaccharides (N-acetyl galactosamine and galactose; GalNAcGal); linear and branched trisaccharides (N-acetyl galactosamine, galactose, and N-acetyl neuraminic acid; GalNAcGalNeuAc); and branched tetrasaccharides (N-acetyl galactosamine, galactose, and two N-acetyl neuraminic acids; GalNAc1Gal1NeuAc2) [69]. A detailed study on the glycosylation of human β-casein has indicated that the C-terminal contains the O-glycan peptide (197-LLNQELLLNPTHQYPVTQPLAPVHNPISV-226) [66], which has antimicrobial functionality [70]. Studies have suggested more extensive glycosylation of κ-casein in camel milk [63] and human milk [71] than in bovine milk. The low percentage of κ-casein and the high level of its glycosylation in camel milk may contribute to its lower pH (6.2–6.7) compared to bovine (6.6–6.7) and human (6.7–6.9) milks [72]. The glycosylation and/or phosphorylation of casein molecules have significant effects on their polarity and functionality.

### 3.4. Predicted Secondary Structures of Milk Caseins in the Four Animals

The biological functions of proteins are determined by two important characteristics: certain amino acid motifs within the sequence, and the three-dimensional conformation of the protein dictating their surface hydrophilicity/hydrophobicity and structure(s) [73]. The amino acid sequence alone is not able to provide all the information hidden in the sequence because certain amino acid alterations have no effects on the biological function(s), while other alterations may have significant effects. Therefore, the three-dimensional structure of a protein is better conserved than its amino acid sequence during evolution due to its tolerance to changes in the primary structure [74].

A two-dimensional structure prediction for the milk caseins of bovines, camels, humans, and African elephants is shown in Figure 6. Milk caseins have been described as rheomorphic (from the Greek rheos meaning stream and morphe meaning form) and are known as “random coil proteins” having less ordered secondary structures and no tertiary structures [75] In general, caseins are characterized by low levels of α-helices in comparison to typical globular proteins. Table 4 shows that all caseins exhibit intrinsic distortedness, which occurs more in αS1- and αS2-caseins than in β- and κ-caseins [76]. Bovine αS-caseins have been reported to have higher predominance of α-helices located close to the sites of phosphorylation. The C-terminal of β-casein and the N-terminal of κ-casein encode high proportions of hydrophobic residues, which limit the formation of α-helices and β-sheets. The high proline content in β-casein is responsible for the formation of long hinges in this casein, which impart surfactant properties to this protein. The possible contribution of proline and glutamine to the secondary structural conformations of αS1-, β-, and κ-caseins has been suggested.

## 4. Casein Micelle Composition and Structure

It has been estimated that a typical bovine milk casein micelle consists of thousands of casein molecules and CCP in a ratio of 94:6 [78]. The phosphoserine residues of the calcium-sensitive caseins associate through their phosphate groups with calcium (or magnesium), which also bind inorganic phosphate, citrate, and water (Figure 7). According to Broyard and Gaucheron [79], different forms of water can be associated with milk caseins as follows: (i) chemically-bound or structural water that is unavailable for chemical reactions; (ii) non-freezable water that is bound to polar amino acids, such as monolayer water; (iii) hydrodynamic hydration water that loosely surrounds proteins; (iv) hydrophobic hydration water that surrounds non-polar amino acids in “cage-like” structures; and (v) capillary water held by surface forces in the proteins. The presence of the different forms of water is responsible for the porous “spongy” structure of casein micelles [80] is formed by cavities (20–30 nm in dimeter) and channels (~5 nm in diameter) [81].

A transmission electron micrograph of a bovine milk casein micelle is shown in Figure 7. Despite wide interest and continuous research on the composition and structure of bovine milk casein micelles, their definite architecture is not fully understood. Several models have been proposed to describe the casein micelle architecture based on physicochemical and microscopic studies [83]. Conventionally, the casein micelles of bovine milk have been described using the progressive “coat-core models” [84], and these models and their extensions have established that κ-casein is distributed throughout the micelle but with a significantly higher concentration at the surface [85]. As discussed below, the presence of κ-casein on the surface confers a net negative charge on micelles, which sterically stabilizes them against flocculation. According to these models, calcium-sensitive caseins are mainly enclosed inside micelles, but their relative distribution and roles are not completely understood. Following the coat-core models are the “submicelle models”, which presume that casein micelles are composed of subunits or submicelles (~12–15 nm) linked together by CCP [85]. The inorganic components (calcium/magnesium and phosphate/citrate) exist in equilibria between casein submicelles and the serum phase [86]. Several internal structure models have been proposed to describe the organization of caseins inside micelles [86]. According to Holt [87], the casein micelle can be viewed as a tangled web of flexible, gel-like protein networks containing microgranules of CCP. Horne [88] proposed the “dual-binding model”, in which caseins associate via hydrophobic regions (rectangular bars) as well as by linkages between their hydrophilic phosphoserine-containing regions and CCP and suggested that high κ-casein concentrations limit the growth of micelles. Further refinement of the internal structure models by Dalgleish [85] led to the proposal of the “mixed channel model” that considers the existence of water cavities. The “calcium phosphate nanocluster model”, the “coat-core model”, emphasizes the flexibility of casein micelles (Figure 7), and it emphasizes the importance of the number and size of phosphorylation clusters as well as their variability within and between animal species [1]. By comparing the amino acid sequences of 20 animal species, ref. [87] suggested that the casein micelle matrix is a dynamic assembly of interchanging structures with different types and degrees of inhomogeneity that are influenced by several environmental factors.

All models agree that the primarily peripheral location of κ-casein is at the surface of the micelle. The “hairy”, negatively charged, glycosylated, hydrophilic glycomacropeptide (GMP, ~7 nm) [5], which protrudes into the aqueous phase, stabilizes micelles and prevents them from aggregating through electrostatic repulsions [5]. The various genetic variants of κ-casein in bovine milk, i.e., A, A^1^, B, C, E, F^1^, F^2^, G^1^, G^2^, H, I, and J, differ slightly in their primary sequence and functional properties, such as gelation potential [30]. In addition to this genetic polymorphism, variants of κ-casein may differ in their degree of glycosylation. Glycosylation of bovine milk caseins is limited to κ-casein, which can have up to nine glycans per molecule, such as galactose, N-acetylgalactosamine, N-acetylneuraminic acid, and sialic acid [89]. It has been suggested that higher concentrations of κ-casein are associated with smaller micelles in bovine milk [85]. However, higher degrees of glycosylation have been shown to be the main reason for smaller micelle sizes in some cow breeds [90]. A previous study on the Montbéliarde cow breed indicated that the B variant of κ-casein exhibits a relatively higher level of glycosylation and smaller average micelle size (~170 nm) than the A variant, which has a larger average casein micelle size (~207 nm). The amino acids in the GMP, representing the hairy part of κ-casein clipped by renins, are highly conserved and mainly negative with only few positively charged conserved residues near the cleavage site [91]. This part of κ-casein plays an important role in maintaining repulsions between micelles and preventing their coagulation in the mammary gland. The hydrophilic amino acid residues in the central region of the ruminant para-κ-casein (PKC) are strongly conserved, which is more important for its function than the conservation of hydrophobic residues [44]. The GMP of all species lacks cysteines, suggesting that cross-linking of this part of κ-casein is unfavorable for the interaction of micelles with the aqueous environment or with each other via disulfide bonds, which may promote their aggregation.

The presence of some levels of β-, αS1-, or αS2-caseins in milk is vital for calcium binding and the micelle internal structure. However, compared to κ-casein, the exact role and the amount of these calcium-binding phosphoproteins necessary for the stabilization of the spatiotemporal organization of the casein micelle remain unknown. This lack of understanding is due to the inter- and intra-differences in the proportions of these proteins in milks from different animal species. The great variation, low sequence identity, and limited conservation of these proteins in different milks (Figure 2 and Table 2) place caseins among the most evolutionarily divergent mammalian proteins [92]. Nonetheless, stable and roughly spherical casein micelles are formed by different mammalian species despite this heterogeneity, suggesting that the constitution of casein micelles follows simple and flexible rules. As the presence of αS-caseins is not obligatory in human and African elephant milks (Table 1), the presence of β-casein is also not vital in milks from knockout mice [93] and in goats that do not express the β-casein gene [93]. Experiments have also shown that approximately 50% of the caseins, predominantly β-casein, dissociate from micelles to the serum phase during cooling of milk [94]. These findings suggest that β- and αS1-caseins may have similar or complementary roles in the constitution and functionality of the casein micelle, a characteristic that may be related to their chaperone activity. The chaperone activity of αS1- and β-caseins, which is critical for the prevention of the aggregation of several proteins (including κ-casein, α-lactalbumin, insulin, lysozyme, alcohol dehydrogenase, and catalase), is facilitated by a higher frequency of proline residues in β-casein (18%) than in αS1-casein (9.2%) [44]. Studies have suggested that there are two types of β-casein that bind in micelles as follows: (i) a portion that is hydrophobically-bound and that can easily be removed by cooling; and (ii) a portion that is tightly bound to CCP nanoclusters [95].

It has been suggested that αS1-casein is required for efficient transport of β- and κ-caseins from the endoplasmic reticulum to the Golgi apparatus in mammary gland epithelial cells and that β- and κ-caseins are retained in the endoplasmic reticulum in αS1-deficient epithelial cells [96]. The lack of αS1-casein in African elephant milk contradicts the generalization of this proposition and supports the need for its modification. Both αS1- and β-caseins are chaperon proteins that establish a dynamic disorder and inhibit the aggregation and formation of amyloid fibrils by κ-casein [96]. Thus, the previous suggestion by le Parc et al. [91] can be modified to state that a certain minimum proportion of αS1- and/or β-casein is needed to ensure casein functionality and maintain the fluidity of milk. Thus, casein micelles can be viewed as dynamic nanoassemblies composed of submicelles linked together by CCP through their phosphoserine residues. In the core of each submicelle, β- and α-caseins are mainly associated via hydrophobic interactions, while κ-casein mainly resides on the surface of micelles, which stabilizes them through electrostatic and steric repulsions [97]. The level of mineralization has been shown to be higher in camel milk (94 mg/g caseins) than in bovine milk (67 mg/g caseins) [98]. Although the calcium and phosphate partitioning between the micellar and serum fractions is almost similar for camel and bovine milks (60–65%), the partitioning of magnesium and citrate substantially differs in casein micelles with 66% and 33% in camel milk compared to 40% and 10% in bovine milk, respectively.

The casein composition of milk critically affects the structure and properties of casein micelles, especially their size and aggregation behavior. As shown in Figure 7, casein micelles vary in size from very small to very large [6] in bovine, camel, and human milks, and they found that the milks are composed of a large number of small particles (40 nm diameter, 80%) and a small number of large micelles with camel milk having a wider distribution and greater number of large particles. The small particles account for 4–8% of the mass or volume of casein in camel milk with an average size of 260–300 nm compared to 100–140 nm in bovine milk. Despite that the low relative κ-casein percentage correlates with a large average casein micelle size [99], the relationship among casein composition, casein micelle size, and dynamics is poorly understood, warranting further investigation of the effect of phosphorylation and glycosylation considering the variability in casein contents and proportions in different animals.

## 5. Functional Properties of Milk Caseins

### 5.1. Milk Types

For health reasons, it is important to understand the different types of milk. A1 β-casein milk is the most abundant milk, it is obtained from cows varieties, such as like Holstein Friesian and Jersey, while A2 β-casein milk is mostly obtained from cows, such as Gir and Sahiwal, and milk from camels and goats [100,101] Compared to A2, A1 milk is relatively cheaper and easier to find. Recently, there has been a difference of opinion about the health effects of the A1 milk type. A1 milk contains BCM-7 (Beta-casomorphine-7) derived from the A1 β-casein during the digestion, which is not present in A2 milk and has been linked to several undesirable health effects [102].

### 5.2. Coagulation Properties

Coagulation and clotting of milk caseins are initiated by either the neutralization of the negative charges of the “hairy” hydrophilic parts of κ-casein by acids or by the cleavage of renin enzymes, leading to destabilization of the milk emulsion and precipitation of caseins [103]. Renin enzymes cleave the Phe^105^-Met^106^ bond of the mature bovine milk κ-casein, producing the hydrophilic GMP (C-terminal amino acid residues 106–169 of the mature bovine κ-casein corresponding to amino acids 127–190 of the whole protein with an approximate molecular weight of 6800 Da) and the hydrophobic PKC (N-terminal amino acid residues 1–105 of the mature bovine κ-casein corresponding to amino acids 22–126 of the whole protein with an approximate molecular weight of 12,000 Da) [26]. In κ-casein, serine occurs at higher levels than threonine in PKC and at lower levels in GMP, suggesting a functional preference for O-glycosylation over phosphorylation in GMP [31]. Furthermore, GMP is unique as it lacks aromatic amino acids and is rich in branched amino acids.

An experimental comparison of camel and bovine milks after they were treated with chymosin, citric acid, and acetic acid for 0, 20, 40, and 60 min is shown in Figure 8 [104]. The coagulation of milk caseins differs markedly between these two milks, with camel milk caseins showing less aggregation and separation of the whey. The acidification of milk starts with the neutralization of aspartate (pKa = 4.25) and glutamate (pKa = 4.05) residues followed by the phosphate groups (pKa = 2.2, 5.8, and 12.4) and sialic acid present as part of O-glycosylation (pKa = 2.2) [105]. This neutralization of the negative charges of the κ-casein hairy layer disables the electrostatic repulsion between casein micelles and allows them to coagulate. The smooth gels of acidified camel milk can be explained by the higher abundance of β-casein (67%) in camel milk compared to that in bovine milk (40%) [106].

The casein composition and micelle structure of milk affect its coagulation and digestibility in the stomach [2], with β-casein having greater effects on digestibility than κ- and α-casein [107,108]. Recently, Zou et al. [109] compared the digestibility of camel, human, and bovine milks by in vitro infant digestion. They observed that human milk does not form a clot in the stomach, however bovine milk forms hard curd, which is in agreement with others [110]. Gastric emptying time in infants, which is shorter for human milk than for bovine milk, correlates with the formation of firm clots of αS1-casein compared to the formation of soft aggregates from β-casein [2]. The nature of the clot formed in the stomach affects the further digestibility of the proteins in the intestine [111]. However, camel milk behaves differently as it instantaneously forms floccules that assemble into a single soft clot of proteins entrapping fat globules [112]. The hardness/softness of the stomach coagulum influences gastric emptying, protein digestion kinetics, and passage through the stomach, and it also affects the digestion in the lower gut and the release of amino acids, especially in individuals with digestive disorders or discomfort [110].

Stomach renins, e.g., chymosin (EC 3.4.23.4) and pepsin (EC 3.4.23.1), hydrolyze κ-casein at low pH values (1.5–2.5), causing the release of GMP into the whey and coagulation of the hydrophobic PKC in the curd [89]. Both chymosin and pepsin clot milk under specific conditions, and they also lead to proteolysis, which ripens the cheese. Chymosin is a principal proteinase that hydrolyzes the Phe-Met bond, and pepsin has a higher and an unspecific proteolytic activity towards peptide bonds, involving aromatic amino acids (Phe, Tyr, and Trp) [89]. The coagulation properties of milk are affected by the κ-casein variants as well as their interaction with the variants of β-casein and β-lactoglobulin [30]. Chymosin-induced coagulation properties of milk are also enhanced by glycosylation [14]. After the initial formation of aggregates by acid, digestion of bovine milk in the stomach proceeds by pepsin and chymosin, leading to the hydrolysis of κ-casein, which occurs at much lower levels in camel and human milks. Bioactive antioxidant, antimicrobial, ACE inhibitory, and dipeptidyl peptidase IV (DPP IV) inhibitory peptides have recently been identified in bovine, camel, and human milks after intestinal digestion [111]. Most of these peptides are derived from β-casein, which is predominant in camel milk. A BLAST search against the UniProt database has indicated that camel β-casein protein shares 69.3% and 61.3% sequence identity with bovine and human β-casein, respectively. It has also recently been shown that camel milk proteins are as equally digestible as their bovine and human counterparts under infant gastrointestinal digestion conditions, and they have been suggested as prospective substitutes to the use in infant formula [112].

More information about the coagulation of milk caseins is available from experimental results on cheese and yogurt quality. These studies are largely available for bovine milk and to some extent for camel milk, but not for human and African elephant milks. Compared to bovine milk, the low κ-casein in camel milk is associated with larger casein micelles, poor milk coagulation properties, and soft/fragile cheese [113]. Similarly, bovine milk with large casein micelles is associated with slow rennet coagulation and reduced cheese firmness [14]. The softness of camel milk cheese is affected by other factors, including high percentage of β-casein and higher proteolytic activity [113]. Further studies on the coagulation properties of milks from different mammalian species may allow identification of the different factors that affect casein coagulation after treatment with renin enzymes and/or with acids.

### 5.3. Ethanol Stability of Milk

In a study by de la Vara et al. [114], ethanol stability of sheep, goat, and cow milk was significantly influenced by pH; a pH increase from 5.7 to 7.1 increased stability significantly. Alhaj et al. [115] reported that camel ethanol stability is affected by sodium chloride (NaCl) concentration and the variation of other minerals that influences the ionic strength. The addition of NaCl to camel milk enhances its sodium and potassium balance, thereby stabilizing the casein micelles.

### 5.4. Heat Sensitivity

Compared to bovine milk caseins, camel milk caseins are less stable at higher temperatures (100–130 °C), and their heat coagulation rate is influenced by pH levels [116,117]. For instance, the coagulation time of camel milk at 130 °C and pH 6.7 is 2–3 min, whereas the coagulation time of bovine milk is 40 min [118]. The reduced level of κ-casein (5% of total casein in camel milk compared with 13.6% in bovine milk) and the absence of β-lactoglobulin may be responsible for the poor stability of camel milk at high temperatures. Camel milk has poor heat stability and cannot be sterilized at natural pH, but the addition of casein and calcium has been found to play a significant role in camel milk heat stability [119].

### 5.5. Proteolysis Sites and Products

Milk caseins are subject to hydrolysis by a wide range of proteolytic enzymes, including endogenous, gastric, and bacterial enzymes, such as pepsin, chymosin, trypsin, chymotrypsin, elastase, carboxypeptidase A, carboxypeptidase B, and leucine amino peptidase (LAP). The open conformation of the caseins explains the ability of proteolytic enzymes to access target bonds and to cause rapid and extensive degradation of casein to smaller peptides [120]. Quadrupole time-of-flight (Q-TOF) mass spectrometry analysis of hydrolytic peptides has discovered that human milk contains several endogenous proteolytic enzymes: plasmin and trypsin (cleave after lysine K and arginine); cathepsin D and elastase (cleave after alanine, isoleucine, leucine, proline, and valine); pepsin and chymotrypsin (cleave after phenylalanine, leucine, tryptophan, and tyrosine); and proline endoperoxidase (cleaves after proline) [121]. Similar enzymes are also present in the gastrointestinal tract of humans and in the milks of other animal species [120].

β-casein is more prone to enzymatic hydrolysis than the other caseins, leading to the release of a large number of peptides. The lack of disulfide bonds and the abundance of proline residues impart an open structure for β-casein that makes it readily available for enzymatic hydrolysis [1]. Plasminogen and its activators, associated with the casein micelle, are mainly responsible for casein degradation by plasmin (EC 3.4.21.7) [121]. The conversion of plasminogen to plasmin is activated by heat treatment during milk processing, and plasmin is heat tolerant and can withstand temperatures as high as those used in ultraheat treatment [122]. Plasmin has a preference to hydrolyze N-terminal Lys-X bonds but is also capable of slowly hydrolyzing Arg-X bonds. Moreover, plasmin readily hydrolyses β- and αS2-caseins as well as slowly hydrolyzes αS1-casein, but k-casein is highly resistant to plasmin [123]. The main proteolytic products produced from β-casein by plasmin include C-terminal peptides [γ1-casein (29–209), γ2-casein (106–209), and γ3-casein (108–209)] and their complementary N-terminal peptides [proteose peptone 8 fast (PP8 fast, 1–28), proteose peptone 8 slow (PP8 slow, 29–105/107), and proteose peptone 5 (PP5, 1–105/107)] [124]. Plasmin has been reported to hydrolyze αS2–casein, resulting in the following 14 fragments: 1–24, 71–80, 115–149, 115–150, 150–165, 151–165, 153–170, 166–173, 167–173, 174–181, 182–188, 182–197, 153–207, and 198–207 [125]. Although plasmin is less active towards αS1-casein, this casein has several cleavage sites where its hydrolysis starts with the cleavage of six fragments (1–22, 91–100, 91–103 103–124, 106–124, and 194–199) from the N-terminal and central parts of the protein, followed by nine other fragments (1–34, 35–90, 80–90, 80–103, 104–124, 104–199, 106–199, 125–199, and 152–199) [126]. Plasmin has also been demonstrated to hydrolyze κ-casein to release the following five fragments: 1–16, 1–24 [127], 1–3, 17–21, and 22–24 [126]. The plasmin hydrolytic peptides of β-casein [128] and κ-casein [129] have been demonstrated to have strong antibacterial properties.

The effect of some of the gastrointestinal enzymes and their combinations on the release of casomorphins from bovine milk β-casein is illustrated in Figure 9. The most well-known of these is β-casomorphin-7 (β-CM-7), which is produced during the digestion of bovine β-casein variants A1 and B with the readily cleavable Ile^66^-His^67^ bond, while variant A2 with the Ile^66^-Pro^67^ bond is resistant to this cleavage [130] The β-CM-7 peptide has been reported to affect numerous opioid receptors in the nervous, endocrine, and immune systems [131]. Other opioid peptides identified in bovine and human milks are listed in Table 5. However, research has shown that bacterial fermentation during yogurt preparation and cheese ripening, as well as renins used in cheese preparation and gastrointestinal tract enzymes, may promote further hydrolysis of opioid peptides and elimination of their harmful effects [132]. For example, lactic acid bacteria have a high intracellular X-prolyl dipeptidyl aminopeptidase (PepX) activity that releases x-Pro dipeptides from the N-terminus of peptides [132]. Bovine BCM-7 has been specifically linked to a higher chance of developing several diseases, including autism, type I diabetes, and cardiovascular diseases [130].

Based on the “leaky gut” theory [135] and the “autism model”, which propose exorphins and serotonin as the important mediators in the development of autism, the effects of β-CM-7 on autism have been studied [136]. With respect to type 1 diabetes mellitus (T1DM), it has been proposed that β-CM-7 may contribute to the impairment of gut-associated immune tolerance and enhance the autoimmune reactions leading to destruction of β cells [127]. Some studies have found an association between an early ingestion of bovine milk by infants and the increased risk for the development of T1DM [137,138]. The opioid activity of human β-CM-7 (Tyr-Pro-Phe-Val-Glu-Pro-Ile) has been found to be 3–30 times lower than that of bovine β-CM-7 (Tyr-Pro-Phe-Pro-Gly-Pro-Ile) according to the guinea pig ileum longitudinal muscle/myenteric plexus preparation assay [139].

Apart from the possible harmful effects of milk opioid peptides, milk hydrolysates and peptides are considered beneficial for health [140]. Table 6 presents some of the bovine casein trypsin-derived peptides with antithrombotic, ACE inhibitory, immunomodulatory, and antimicrobial activities.

### 5.6. Chaperone Activities

Molecular chaperons stabilize other proteins against unfolding, aggregation, and precipitation under conditions of thermal and other environmental stress situations. The mechanism of chaperone action is not fully understood, but researchers have suggested that it involves hydrophobic interactions and complex formations with partially folded proteins to enable their solubilization through hydrophilic regions. The less ordered and non-compact secondary and tertiary structures of caseins enable them to possess chaperone activity through their hydrophobic and hydrophilic regions (Figure 4). The chaperone activity is responsible for the solubilization of target proteins and the prevention of their aggregation and fibrillation. For example, it has been observed that αS1-casein, but not β- and κ-caseins, prevent the aggregation and precipitation of reduced insulin and α-lactalbumin [45]. αS2- and κ-caseins have minimal chaperone activity and can form amyloid fibrils, which are inhibited by αS1- and β-caseins [147]. Dissociation of κ-casein from casein micelles is a prerequisite for amyloid fibril formation, and oxidation of methionine residues enhances their formation [147]. The large number of proline residues, the extent of exposed hydrophobic surfaces, the polar phosphorylated residues, and the N-terminal hydrophilic domain may explain the chaperone activity of β-casein [148]. Phosphorylation of αS1- and β-casein plays an important role in the chaperone activity of these caseins [149]. Bovine β-casein exhibits higher chaperone activity against alcohol dehydrogenase aggregation than camel β-casein, which may be due to a higher net charge and a greater surface hydrophobicity of bovine β-casein, leading to stronger amphiphilicity and “detergent” properties [150]. The chaperone-like activities of milk caseins are important not only for their biological functions but are also expected to play a role during food processing, e.g., the stability of ultrahigh temperature (UHT) milk [4].

### 5.7. Nanoencapsulation Properties

Casein micelles, sized 50 to 500 nm, are natural carriers of calcium and phosphate, and they have great potential to serve as nanoencapsulation carriers for a variety of hydrophilic and hydrophobic bioactive components [151]. Examples of bioactive compounds that have been encapsulated in reassembled casein micelles or sodium caseinates include ω-3 polyunsaturated fatty acids [152], vitamin D2, vitamin D3 [153], vitamin E, β-carotene [154], catechins [154], quercetin [151], folic acid [155], and anticancer drugs [156]. Casein micelles are stable during food processing, safe, protective against oxidation, and highly bioavailable. Thus, the diversity of casein structures present in the milk of different animal species will allow for the effective and tailored design for encapsulation of different bioactive nutraceuticals [12]

## 6. Nutritional Properties and Applications

Casein polymorphism is influenced the composition, texture, and functional properties of milk products [157]. Studies have shown that casein polymorphism has been associated with ischemic heart disease, cardiovascular disease, type 1 diabetes, sudden infant death syndrome, neurological disorders, such as autism and schizophrenia, lactose intolerance, and various allergies [158,159]. Therefore, careful attention should be paid to casein polymorphism, and deeper research is needed to verify the range and nature of its interactions with the human body, especially in the gastrointestinal tract. In this review our focus would be limited on diabetes, allergies, and autism.

### 6.1. Diabetes

According to the International Diabetes Federation (IDF), approximately 10% of the world population is living with diabetes [160]. T1DM, which accounts for 10% of the total cases, is an immune disease characterized by the death of pancreatic β cells and the loss of their ability to secrete insulin [160]. This disease has a genetic component but is triggered by diet and other environmental factors [161,162]. Genetically predisposed T1DM individuals are known to exhibit intestinal barrier abnormalities that permit the exposure of the intestinal immune system to dietary antigens, such as proteins and peptides, leading to immune activation and intestinal inflammation. Figure 10 presents the association between T1DM and bovine milk A1 β-casein, which has been proposed as an important trigger, and explains the differences in T1DM prevalence in different countries [137]. It has been suggested that milk from dairy cow breeds rich in the A1 genetic variant of β-casein, but not in the A2 variant, is associated with incidences of T1DM due to the generation of the immunomodulatory and diabetogenic peptide BCM-7 [163]. The association between bovine milk and T1DM is explained by immunological cross-reactivity (molecular mimicry) between milk proteins and islet autoantigens [164]. The American Academy of Pediatrics has recommended that infants, especially from families with history of T1DM, be breast fed and not subjected to bovine milk feeding at an earlier stage [165]. Further studies are needed to investigate the levels of A1/B β-casein and the age of exposure that can lead to etiology of T1DM in children.

In type 2 diabetes (T2DM), which accounts for 90% of the total world diabetes cases [160], pancreatic β cells produce insulin, but the utilization of insulin by cells is compromised due to inefficient insulin receptors on cell membranes. Glucose transporters have a central role in glucose sensing, glucose homeostasis, and T2DM. Once insulin binds to the insulin receptor (IR), GLUT4 translocates to the membrane and increases glucose transportation into the cell, and the insulin receptor substrate 1 (IRS-1) is activated by phosphorylation [166]. This in turn activates phosphoinositide 3-kinase (PI3K) and protein kinase B (AKT), which initiate GLUT4 translocation. Any defect in the IR and insulin signaling negatively affects the translocation of GLUT4 and increases blood glucose concentration in T2DM patients [166]. The link between milk protein consumption and T2DM is less obvious than with T1DM. Several meta-analyses of observational studies have found an inverse association of bovine milk and dairy product consumption with T2DM [167,168]. The evidence is stronger for the association of low incidence of T2DM and consumption of fermented dairy products, especially yogurt, than for pasteurized milk [169,170]. As with T1DM, the protective milk components and the mechanism(s) involved in this protection are unknown. However, some proteolysis products from milk caseins may be involved in this effect.

As evident from some studies, [171,172], including observational and human intervention research as well as animal intervention studies and in vitro cell studies, camel milk has been demonstrated to have an anti-diabetic impact against both T1DM and T2DM. The first discovery of the antidiabetic potential of camel milk was made by the observation that the Raica community in India has zero prevalence of diabetes compared to neighboring control groups, and this difference is attributed to the consumption of camel milk by this nomadic group [173]. The Raica community has also been shown to have a high prevalence of diabetes susceptibility genes, highlighting the importance for identifying the protective role of camel milk and other dietary and life-style factors [174]. Many intervention studies on humans and animals summarized in systematic reviews [175,176] have confirmed the epidemiological observations, suggesting that camel milk protects against T2DM and T1DM. Compared to bovine milk, camel milk contains key pharmacological molecules, including lactoferrin, lactoperoxidase, and peptidoglycan recognition protein, which have been shown to reduce blood sugar, decrease insulin resistance, and improve lipid profiles [177,178]. It has been suggested that daily consumption of 500 mL of camel milk causes a significant decrease in fasting blood glucose, fasting HbA_1c_, and plasma insulin levels in both T1DM and T2DM [178]. A small study involving children has reported similar but not significant results [177]. The actual antidiabetic agent(s) in camel milk and their mechanisms of action are still not known. The hypoglycemic properties of camel milk have been ascribed to an insulin-potentiating peptide in fresh milk with a molecular weight less than 10 kDa that acts allosterically at the level of the IR by differentially impacting its downstream signaling [179]. Camel milk contains significantly higher levels of insulin (18–55 mIU/L) compared to bovine milk (ca. 17 mIU/L) [180]. Although camel and bovine insulin proteins are topologically similar, it is possible that insulin in camel milk is protected by encapsulation in protein nanoparticles or lipid microvesicles, which allows it to resist stomach digestion and reach the bloodstream [181]. The functional differences between camel and bovine milk may also be attributed to their sequence divergence as well as whey proteins. The predominant casein in camel milk (containing 65% of the total caseins in camel milk as opposed to 40% in bovine milk) is substantially distinct from the bovine β-casein variants [106]. It has been reported that dipeptidyl peptidase IV (DPP-IV) inhibitory peptides exist in camel milk protein hydrolysates [182]. DPP-IV inhibitory peptides with molecular masses less than 2 kDa [183] encode many hydrophobic amino acid residues [184]. However, it remains unknown if the active antidiabetic agent in camel milk belongs to caseins, whey proteins, or both. Studies are being conducted to address this question.

### 6.2. Allergy

Cow milk allergy (CMA) occurs in infants and young children due to immunologically adverse reaction(s) to one or several milk proteins [185]. The prevalence of CMA is estimated at 0.1–0.5% in adults and at 2–6% in infants because their guts are not sufficiently developed to digest and immunologically adapt to the high intake of milk proteins, which are the first encountered antigens [186]. CMA is a complex IgE-/non-IgE-mediated disorder manifested at the level of digestive tract (50–60%), skin (50–60%), and/or respiratory tract (20–30%), and it ranges in severity from mild to severe [187]. In addition to whey proteins, the four bovine milk caseins (αS1-, αS2-, β-, and κ-) have been implicated in CMA (Table 7). As evident from this data, CMA is a complicated multimodal allergy because it is not restricted to one protein, and several epitopes are involved within each protein. All bovine milk caseins together and the individual caseins have been demonstrated to be potent allergens in predisposed individuals, inducing marked IgE responses, which suggests that both distinct and common epitopes are involved [188]. The common epitopes involve CPPs with the SerP-SerP-SerP-Glu-Glu motif, corresponding to sequences 66–70 of αS1-casein and 17–21 of β-casein, as well as 8–12 and 56–60 of αs2-casein [189]. The immunoreactivity of CPPs results from their liberation during intestinal proteolysis and their resistance to further degradation by digestive enzymes [190]. Reduction of CMA can be achieved by hydrolysis of milk, but this is challenged by bitterness due to extensive hydrolysis and by residual allergens due to mild hydrolysis.

The cross-reactivity between human and non-human milk proteins is related to the extent of dissimilarity with human proteins [192]. While bovine, goat, and sheep milk caseins share a significant dissimilarity, camel and horse milk caseins are more homologous to human milk caseins, which may explain their weak cross-reactivity and lower allergenicity [193]. Thus, milks from other animal species are being considered as alternatives for bovine milk, especially for infants and young children. After shown to be tolerable by a supervised oral challenge test, donkey milk, which is more similar to human milk than bovine milk, has been suggested as a substitute for bovine milk for children with severe IgE-mediated CMA; however, there is a limited supply of donkey milk [194]. A previous study on 38 children suffering from CMA has shown that 18.4% cross-sensitization is observed for camel milk compared to 63.2% with goat milk, suggesting that camel milk is a better alternative than cow milk [195]. The matrix-assisted laser desorption/ionization-time of flight (MALDI-TOF) mass spectrometry to compare human milk to other mammalian (cow, buffalo, goat, sheep, donkey, and camel) milks found that camel milk has a unique spectral profile that supports the nutritional needs of children [196]. Comparing the allergenicity of camel and bovine milk caseins and whey proteins in brown Norway rats, and they found that both camel and bovine milks contain similar immunogenic and allergenic activities, as evident from the induction of comparable eliciting capacities and levels of specific IgG1 and IgE antibodies. However, a low cross-reactivity is observed between camel and bovine proteins, and the cross-reactivity is even lower between caseins than between whey proteins [197]. It has been suggested that caseins possess epitopes of the linear type, while whey proteins are dominated by conformational epitopes. Studies have confirmed that camel milk has lower allergenicity than bovine milk, and that it can be used in child nutrition after a negative skin prick test and low IgE titers [193,198]. The allergenicity of milk caseins is also affected by their digestibility. For example, the slightly lower allergenicity of goat milk compared to bovine milk attributes to the lack of αs1-casein, causing a decrease in the digestibility of allergic β-lactoglobulin [199].

### 6.3. Autism

Autistic spectrum disorder (ASD) is a neurodevelopmental disorder marked by behavioral and communication deficits, as well as restricted interests and repetitive behaviors [200]. According to the World Health Organization (WHO), it is estimated that the worldwide prevalence of ASD is 0.76 percent [201], and according to the Centers for Disease Control and Prevention (CDC), 1 in 59 children aged 8 years in the United States has ASD [202]. Autism is caused by both genetic and environmental factors that affect the developing brain [200]. The incidence of autism and its relationship with milk casein have been reported [131]. Consumption of A1 β-casein has been found to be related to several neurological disorders, including autism, as hydrolysis of β-casein produces the BCM-7 causative agent [131]. BCM-7, which has morphine-like activity, has been found to significantly contribute to autism incidence. Patients with autism have elevated levels of BCM-7 both in their urine and blood [135]. However, consumption of A2 β-casein has been found to decrease the risk of coronary heart disease and some neurological disorders, such as autism, as well as increase the level of low-density lipoprotein (LDL) and high-density lipoprotein (HDL) cholesterol [203,204].

## 7. Conclusions

This review discusses caseins in cow, camel, human, and African elephant milks to highlight multiple sequence and structural differences among mammalian caseins. Bovine and camel milks contain αS1-, αS2-, β-, and κ-caseins at different relative ratios (38:10:40:12 and 22:9:65.5:3.5, respectively). Moreover, human milk lacks αS2-casein, and African elephant milk lacks both αS1- and αS2-caseins (Table 1). This variability supports the suggestion that only κ-casein and an ancient β-casein-like protein are obligatory to form stable casein micelles, and αS1- and αS2-caseins have developed in some species later during evolution. The exact roles that the different caseins play in the stabilization of micelles and the properties of milk are still not fully understood. In general, the level of diversity in caseins results in their functional properties and biological roles. Several models have described the internal structure of casein micelles, in which κ-casein is thought to concentrate on the surface. One discretion is that casein molecules fill the matrix of casein micelles, in which CCP nanoclusters are dispersed and bind casein molecules together. It has been proposed that calcium phosphate clusters are bridged by hydrophobic interactions and cross-linked to form casein supramolecules. Understanding casein micelle structures and the role of different caseins in the stabilization of micelle structures will help realize their effects on the functional and nutritional values of different milks to facilitate the development of nutritionally improved and readily digestible food products. Also, careful evaluation of the casein polymorphism is required, as it relates to nutrition, function, and health. A deeper understanding of its interactions with the human gastrointestinal tract would be beneficial.

## Figures and Tables

**Figure 1 molecules-28-02023-f001:**
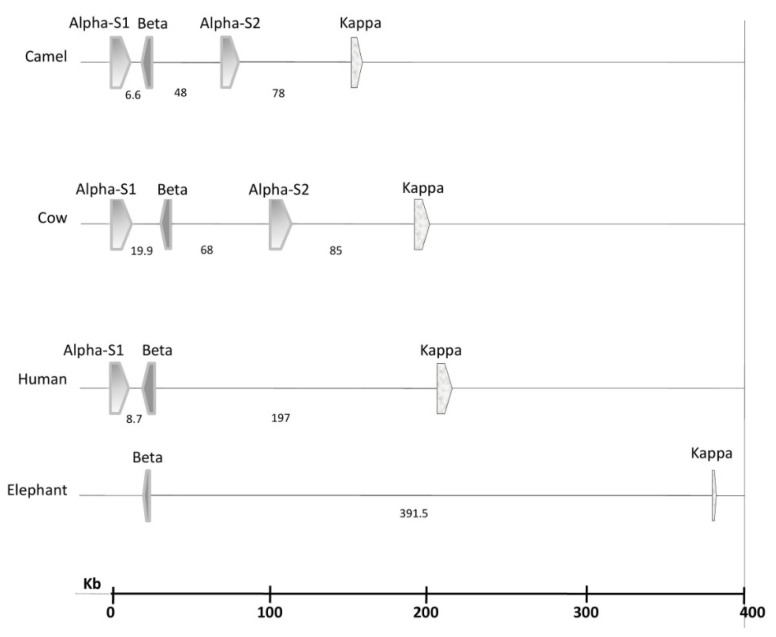
Multispecies comparison of the casein gene loci cluster, illustrating the distances between the casein genes within the four animal species. Data source [19].

**Figure 2 molecules-28-02023-f002:**
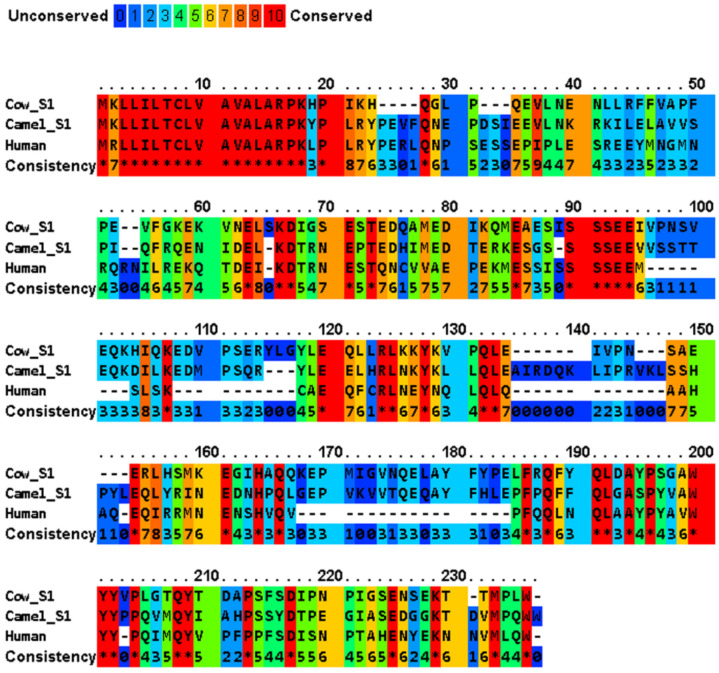
Multiple sequence alignment of cow milk caseins (αS1- variant B, αS2- variant A, β-variant A1, and κ-variant A) and those of the major camel, human, and elephant caseins, and indication of conserved and unconserved amino acid residues. The amino acid sequences are retrieved from UniProt/NCBI database, and the comparison of protein sequences was performed using PRALINE server (https://www.ibi.vu.nl/programs/pralinewww/, accessed on 13 June 2022) to localize the conserved sequence segments. As the selected sequences may show homoplasy, a low E-value inclusion threshold of 0.001 is used to build the alignment through 10 iterative position specific PSI-BLAST iterations. Amino acid residues: alanine (A), asparagine (N), aspartic acid (D), cysteine (C), glutamic acid (E), phenylalanine (F), glycine (G), histidine (H), isoleucine (I), lysine (K), leucine (L), methionine (M), proline (P), glutamine (Q), arginine (R), serine (S), threonine (T), valine (V), tryptophan (W), and tyrosine (Y) [34]. The amino acid sequences of the signal peptides of the different proteins are shown in Table 3.

**Figure 3 molecules-28-02023-f003:**
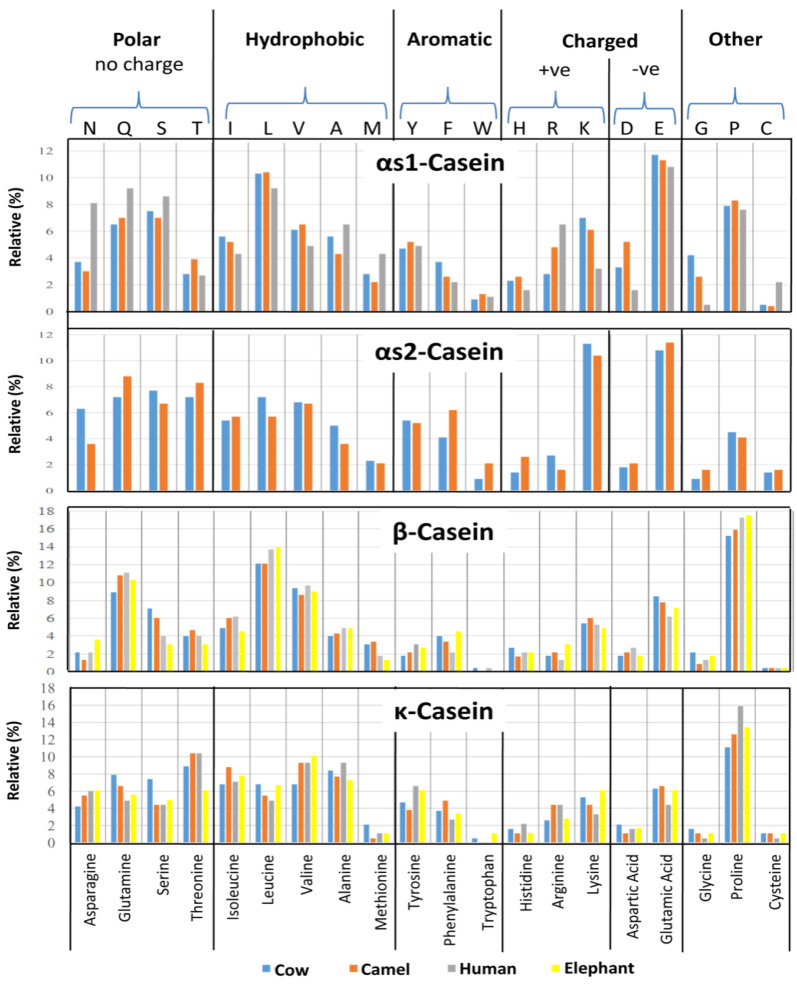
The percentage distribution of the amino acid residues of αS1-, αS2-, β-, and κ-caseins of the milks of cow, camel, human, and elephant.

**Figure 4 molecules-28-02023-f004:**
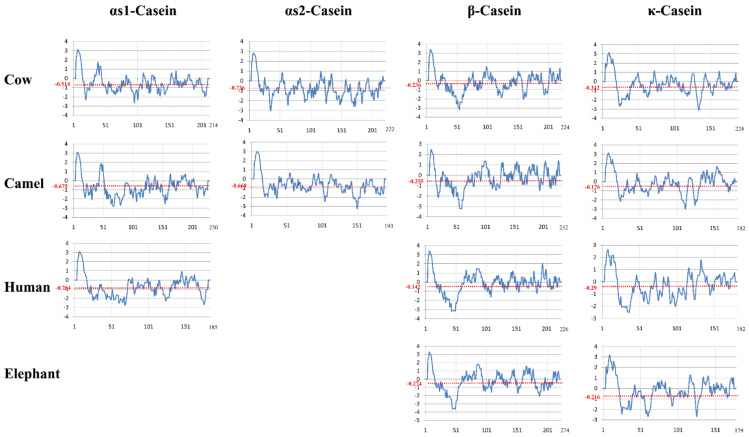
Scoring variation of the hydropathy index for the αs1-, αs2-, β-, and k-caseins of the milks of cow, camel, human, and elephant calculated with a default window size of 9 amino acid residues on the basis of the [51] parameters through Pepinfomodule of EMBOSS [52].

**Figure 5 molecules-28-02023-f005:**
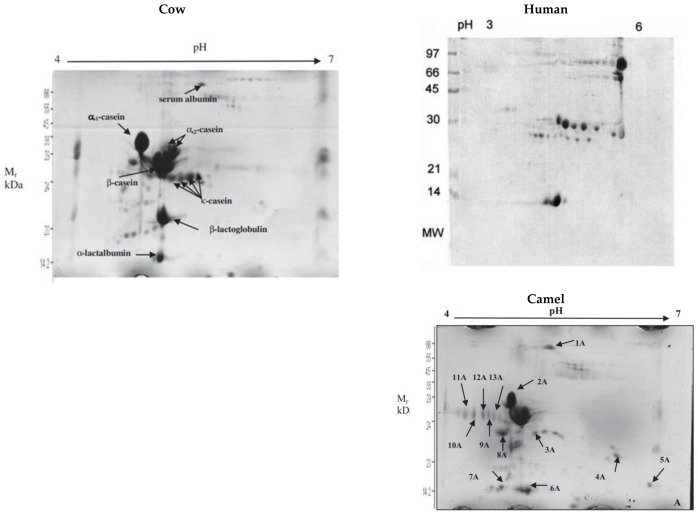
Two-dimensional gel electropherograms showing the different caseins in cow, human, and camel milks. For camel milk: spots 2A and 8A (aS1-casein), spots 3A, 4A, 5A, and 7A (b-casein), and spots 9A, 10A, 11A, 12A, and 13A (k-casein). Source: [65] eproduced with Creative Commons permission.

**Figure 6 molecules-28-02023-f006:**
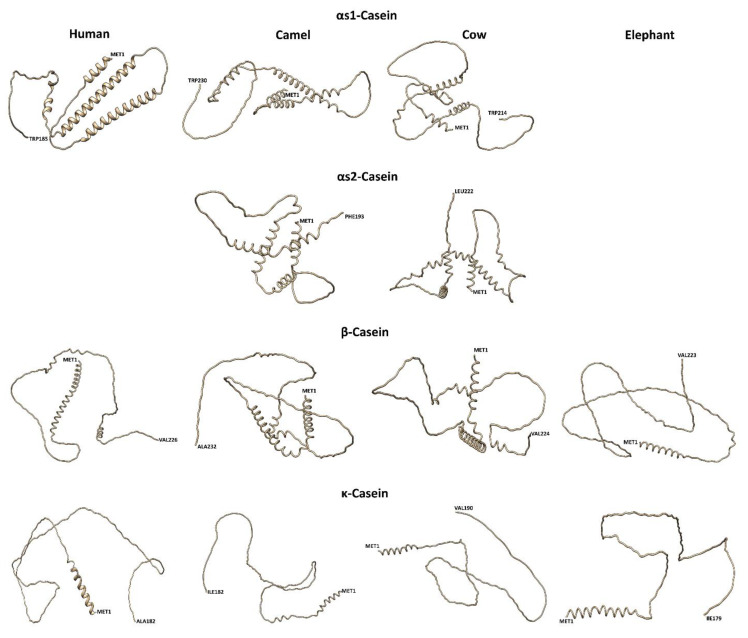
Predicted casein models are visualized using the silhouette and licorice presets of Chimera, using the RaptorX server. As the predicted data is unable to show the real topological scenario and as the caseins naturally encode a substantially low fraction of secondary structures, the casein models were predicted through the current most accurate Alphafold methodology. (https://colab.research.google.com/github/sokrypton/ColabFold/blob/main/AlphaFold2.ipynb, accessed on 13 June 2022) on the basis of multiple sequence alignments, generated by MMSeqs2 [77].

**Figure 7 molecules-28-02023-f007:**
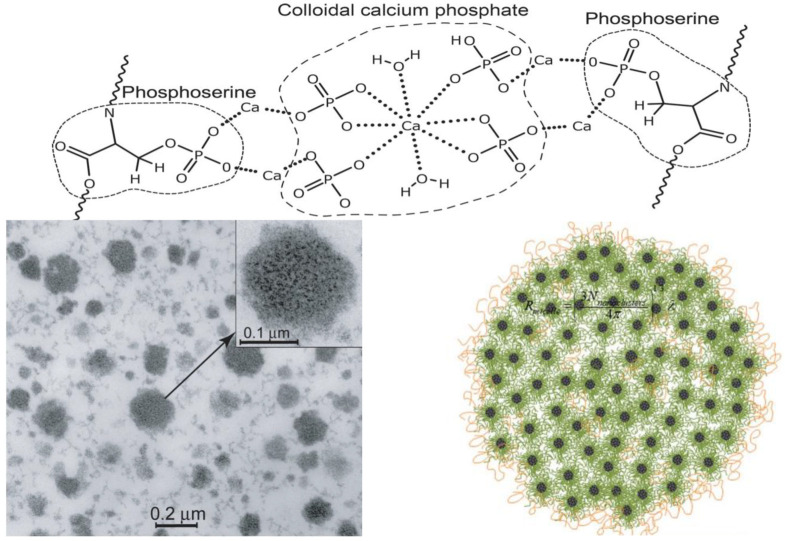
Postulated structure of casein micelle inter-casein linkages of phosphoserine residues to colloidal calcium phosphate. Sources: [5,82]. Reproduced with permission from Elsevier.

**Figure 8 molecules-28-02023-f008:**
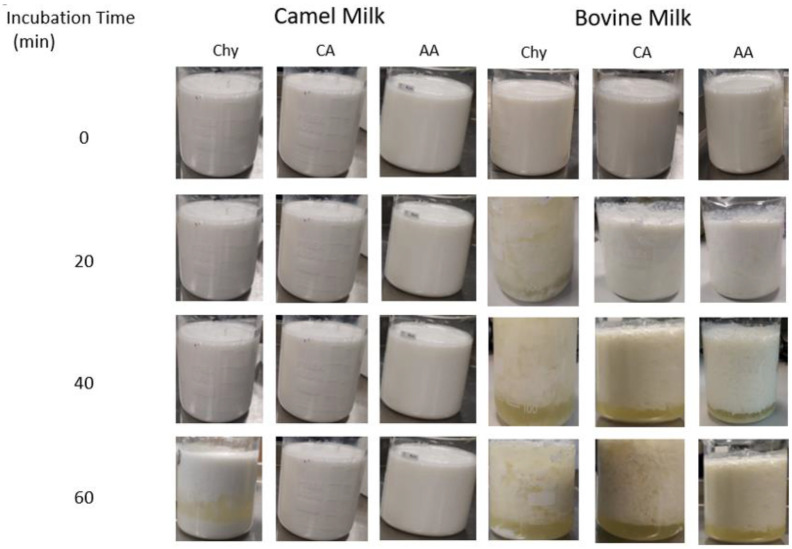
Photographed images of coagulation behaviors with time (0–60 min). Chy (chymosin), CA (citric acid), AA (acetic acid). Source: [104] Reproduced with permission from Elsevier.

**Figure 9 molecules-28-02023-f009:**
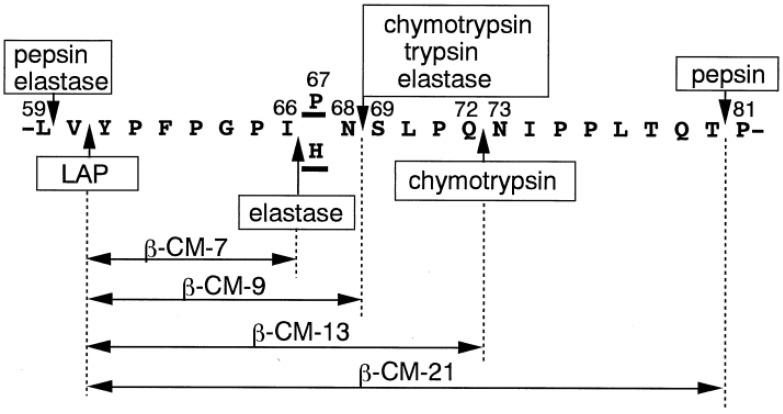
Selected enzymes and enzyme combinations involved in the proteolysis of cow β-casein leading to the release of β-casomorphins (β-CM). Source: [133]. Reproduced with permission from Elsevier.

**Figure 10 molecules-28-02023-f010:**
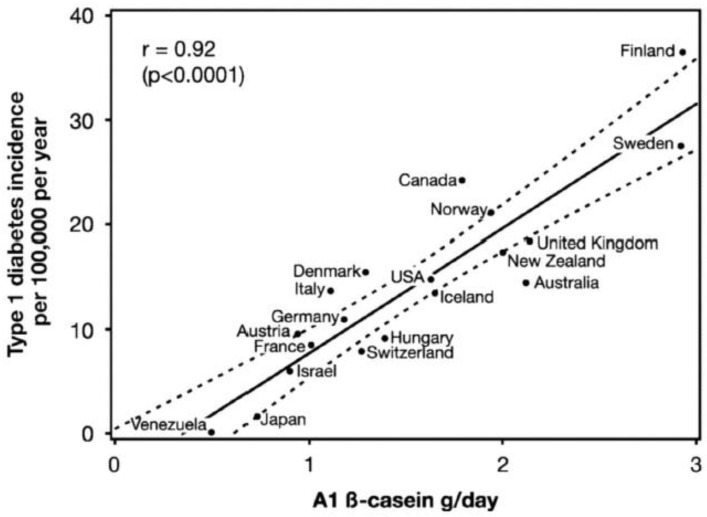
Correlation between A1 β-casein supply per capita in 1990 and type 1 diabetes incidence (1990–1994) in children aged 0–14 years in 19 countries. (r = 0.92; 95% confidence interval: 0.72–0.97; *p* < 0.0001). Dotted lines are the 95% confidence limits of the regression line. Reproduced from [163] Creative Commons permission.

**Table 1 molecules-28-02023-t001:** Casein composition and micelle size of milks from humans and selected mammalian species *.

Animal	Total Casein Concentration (g/L Milk)	Relative Casein Composition (%)					Mean Micelle Diameter (nm)
		αS1	αS2	β	κ	αS1 + β	
African elephant	Unknown	-	-	89	11	89	350–700
Buffalo	32–40	40	9	35	12	75	190
Cow	24.6–28	38	10	40	12	78	150–182
Camel	22.1–26.0	22	9	65.5	3.5	87	380
Goat	23.3–46.3	20	16	41	17	61	180–300
Horse	9.4–13.6	17.7	1.5	79	1.8	96	255
Human	2.4–4.2	3	-	70	27	73	64–80
Sheep	41.8–52.6	50	-	40	10	90	180–210
Range	-	3–55	9–28	26–89	3.5–27	61–96	-

* Data was adapted from [2] except for African elephant [9].

**Table 2 molecules-28-02023-t002:** Sequence identity between the αs1-, αs2-, β- and κ-caseins of the studied milks including the signal peptides (%) *.

Alpha-s1	Cow	Camel		Alpha-s2	Cow		
Camel	50.24			Camel	63.16		
Human	39.66	47.75					
beta-	Cow	Camel	Human	kappa-	Cow	Camel	Human
Camel	66.82			Camel	60.22		
Human	56.07	62.9		Human	54.14	59.34	
Elephant	54.76	55.3	62.61	Elephant	48.31	56.25	53.98

* Sequence similarity among the subset members scrutinizing through CLUSTAL W [35]. Elephant milk lacks alpha-s1 & alpha-s2 caseins. Human milk lacks αs2 casein (See Table 1). Data source [36,37].

**Table 3 molecules-28-02023-t003:** Key physicochemical parameters showing the diversity in isoelectric points (pI) and stability scores of the different caseins as estimated from the sequence dataset through ProtParam (https://web.expasy.org/protparam/, accessed on 13 June 2022).

Animal		Accession Number	Whole Protein				Mature Protein *				Signal Peptide (retrieved from the Uniprot Database (The UniProt Consortium, 2017)
			Sequence Length	Molecular Weight (KDa)	Theoretical pI	Instability Index	Sequence Length	Molecular Weight (KDa)	Theoretical pI	Instability Index	
αS1-Casein											
1	Cow	P02662	214	24.5289	4.98	56.03	199	22.9748	4.91	57.99	MKLLILTCLVAVALA
2	Camel	O97943	230	26.8614	4.96	64.07	215	25.3073	4.89	66.44	MKLLILTCLVAVALA
3	Human	P47710	185	21.6710	5.32	70.90	170	20.0894	5.17	74.75	MRLLILTCLVAVALA
αS2-Casein											
4	Cow	P02663	222	26.0187	8.55	44.68	207	24.3485	8.34	46.27	MKFFIFTCLLAVALA
5	Camel	O97944	193	22.9641	6.00	58.11	178	21.2659	5.80	61.08	MKFFIFTCLLAVVLA
β-Casein											
6	Cow	XP_010804480.2	259	29.22125	6.17	90.11	209	23.6924	5.38	97.56	MKVLILACRVALALA
7	Camel	Q9TVD0	232	26.2178	5.62	96.58	217	24.6507	5.27	100.04	MKVLILACRVALALA
8	Human	P05814	226	25.3810	5.52	73.18	211	23.8578	5.33	74.17	MKVLILACLVALALA
9	Elephant	G3TDS4	223	25.2770	6.21	72.14	208	23.6995	5.97	73.06	MKVFILACLVAFALG
κ-Casein											
10	Cow	P02668	190	21.2693	6.29	54.21	169	18.9744	5.93	57.46	MMKSFFLVVTILALTLPFLGA
11	Camel	P79139	182	20.4176	8.55	44.72	162	18.2538	8.03	46.49	MKSFFLVVTILALTLPFLGA
12	Human	P07498	182	20.3010	8.97	56.48	162	18.1628	8.68	59.84	MKSFLLVVNALALTLPFLAV
13	Elephant	XP_023408995.1	181	20.5520	8.45	48.56	161	18.3880	7.98	46.37	MKGFLLVVNILLLPLPFLAA

* Elephant milk lacks αS1 and αS2 caseins. Human milk lacks αS2 casein. Physicochemical parameters were estimated according to the method of [59]. Instability index lower than 40 affirms the stability of a protein [60].

**Table 4 molecules-28-02023-t004:** The percentages of α-helixes, β-sheets, coils, and disordered residues in αs1-, αs2-, β- and κ-caseins of the studied animals (%) *.

	Animal	Accession Number	Percentage Conformation (%)			Disorder (%)
			α-Helixes/Exposed	β-Sheets/Medium	β-Turns/Buried	
αS1-Casein						
1	Cow	P02662	38/64	1/14	59/21	68
2	Camel	O97943	36/58	3/17	59/23	61
3	Human	P47710	46/66	3/12	49/20	69
αS2-Casein						
4	Cow	P02663	48/70	0/14	50/14	57
5	Camel	O97944	43/58	0/19	56/22	58
β-Casein						
6	Cow	XP_010804480.2	29/64	2/20	68/15	46
7	Camel	Q9TVD0	26/64	1/22	71/13	46
8	Human	P05814	23/57	1/27	75/14	38
9	Elephant	G3TDS4	19/60	1/23	78/15	41
κ-Casein						
10	Cow	P02668	8/58	5/19	86/21	53
11	Camel	P79139	6/52	7/19	86/28	37
12	Human	P07498	6/48	6/24	86/26	26
13	Elephant	XP_023408995.1	14/56	7/20	77/23	30

* Elephant milk lacks αS1 & αS2 caseins and human milk lacks αS2 casein. The intrinsically disordered residues are screened to analyze the extent of the rheomorphic characteristics for the different casein molecules across the selected species.

**Table 5 molecules-28-02023-t005:** Opioid peptides reported to be derived from β-, α-, and αS1-caseins of cow and human milk.

Animal	Opioid Peptide	Amino Acid Sequence
Cow	β-casomorphin-4	Tyr^60^-Pro-Phe-Pro^63^ (YPFP)
	β-casomorphin-5	Tyr^60^-Pro-Phe-Pro-Gly^64^ (YPFPG)
	β-casomorphin-6	Tyr^60^-Pro-Phe-Pro-Gly-Pro^65^ (YPFPGP)
	β-casomorphin-7	Tyr^60^-Pro-Phe-Pro-Gly-Pro-Ile^66^ (YPFPGPI)
	β-casomorphin-8	Tyr^60^-Pro-Phe-Pro-Gly-Pro-Ile-Pro/His^67^ (YPFPGPIP/H)
	β-casomorphin-11	Tyr^60^-Pro-Phe-Pro-Gly-Pro-Ile-Pro-Asn-Ser-Leu^70^ (YPFPGPIPNSL)
	k-casein-casoxin-A	Tyr^35^-Pro-Ser-Tyr-Gly-Leu-Asn-Tyr^42^ (YPSYGLNY)
	k-casein-casoxin-B	Tyr^58^-Pro-Tyr-Tyr^61^ (YPYY)
	k-casein-casoxin-C	Tyr^25^-Ile-Pro-Ile-Gln-Tyr-Val-Leu-Ser-Arg^34^ (YIPIQYVLSR)
	αS1-casein-exorphin	Arg^90^-Tyr-Leu-Gly-Tyr-Leu-Glu^96^ (RYLGYLE)
Human	β-casorphin-4	Tyr^41^-Pro-Ser-Phe4^44^ (YPSF)
	β-casomorphin-4	Tyr^51^-Pro-Phe-Val^54^ (YPFV)
	β-casomorphin-5	Tyr^51^-Pro-Phe-Val-Glu5^55^ (YPFVE)
	β-casomorphin-7	Tyr^51^-Pro-Phe-Val-Glu-Pro-Ile^57^ (YPFVEPI)
	β-casomorphin-8	Tyr^51^-Pro-Phe-Val-Glu-Pro-Ile-Pro^58^ (YPFVEPIP)
	k-casein-casoxin-D	Tyr^158^-Val-Pro-Phe-Pro-Pro-Phe^164^ (YVPFPPF)

Data was adapted from [134]. Reproduced with permission from Elsevier.

**Table 6 molecules-28-02023-t006:** Selected bioactive peptides from the proteolysis of cow milk caseins, camel and human (black cow, red camel, and blue human) *.

Casein	Peptide	Bioactivity
κ-Casein	^103^LSFMAIPPK^111^	Antithrombotic
	^108^IPP^110^	ACE-inhibitory
	^106^MAIPPKK^112^	Antithrombotic
	^106^MAIPPKKNQDK^116^	Antithrombotic
	^113^NQDK^116^	Antithrombotic
β-Casein	^1^RELEELNVPGEIVESLSSSEESITR^25^	Calcium-binding
	^1^RELEELNVPGEIVESLSSSEESITRINK^28^	Immunomodulatory, calcium-binding
	^2^ELEELNVPGEIVESLSSSEESITRINK^28^	Calcium-binding
	^63^PGPIPN^68^	Immunomodulatory
	^191^LLY^193^	Immunomodulatory
	^193^LLYQEPVLGPVRGPFPIIV^209^	Immunomodulatory
	^74^IPP^76^	ACE-inhibitory
	^84^VPP^86^	ACE-inhibitory
	^108^EMPFPK^113^	ACE-inhibitory
	^177^AVPYPQR^183^	ACE-inhibitory
	^193^YQEPVL^198^	ACE-inhibitory
	^193^YQEPVLGPVRGPFPI^202^	ACE-inhibitory
	^193^YQEPVLGPVRGPFPIIV^209^	Antimicrobail
	^199^GPVRGPFPIIV^204^	ACE-inhibitory
αS1-Casein	^1^RPKHPIKHQGLPQEVLNENLLRF^23^	Immunomodulatory, antimicrobial
	^194^TTMPLW^199^	Immunomodulatory
	^23^FF^24^	ACE-inhibitory
	^23^FFVAP^27^	ACE-inhibitory
	^43^DIGSESTEDQAMEDIK^58^	Calcium-binding
	^45^GSESTEDQAME^55^	Calcium-binding
	^59^QMEAESISSSEEIVPNSVEQK^79^	Calcium-binding
	^66^SSSEEIVPN^74^	Calcium-binding
	^102^KKYKVPQ^109^	ACE-inhibitory
	^106^VPQLEIVPNSAEER^119^	Calcium-binding
	^142^LAYFYP^147^	ACE-inhibitory
	^157^DAYPSGAW^164^	ACE-inhibitory
	^194^TTMPLW^199^	ACE-inhibitory
αS2-Casein	^1^KNTMEHVSSSEESIISQETYKQEKNMAINPSK^32^	Immunomodulatory
	^2^NTMEHVSSSEESIISQETYK^21^	Calcium-binding
	^46^NANEEEYSIGSSSEESAEVATEEVK^70^	Calcium-binding
	^55^GSSSEESAEVATEEVKITVDD^75^	Calcium-binding
	^126^EQLSTSEENSK^136^	Calcium-binding
	^138^TVDMESTEVFTK^149^	Calcium-binding
	^164^LKKISQRYQKFALPQY^179^	Antimicrobial
	^165^KKISQRYQKFALPQYLKTVYQHQKAMKPWIQPKTKVIPY^203^	Antimicrobial
	^174^FALPQY^179^	ACE-inhibitory
	^174^FALPQYLK^181^	ACE-inhibitory
	^183^VYQHQKAMKPWIQPKTKVIPYVRYL^207^	Antimicrobial
	^189^AMKPW^193^	ACE-inhibitory
	^189^AMKPWIQPK^197^	ACE-inhibitory
	^190^MKPWIQPK^197^	ACE-inhibitory
	^198^TKVIP^202^	ACE-inhibitory
Camel αs1-CN	^4^DNLMPQFM^8^	DPP-IV binding
	^1^WNWGWLLWQL^9^	DPP-IV binding
	^1^TF^2^	ACE-inhibitory
	1LxV3	Glycose intake
Human αs1-CN	^1^YPER^18^	ACE-inhibitory
	^136^YYPQIMQY^143^	ACE-inhibitory
	^164^NNVMLQW^170^	ACE-inhibitory
Human β-CN	^169^VPYPQ^173^	Antioxidant
	^154^WSVPQPK^160^	Antioxidant
	^54^VEPIPY^59^	Immunostimulatin
Human κ-CN	^31^YPNSYP^36^	Antioxidant
	^53^NPYVPR^58^	Antioxidant

* Source: [141,142,143,144,145,146] Reproduced with permission from Elsevier.

**Table 7 molecules-28-02023-t007:** The identified allergic epitopes of the major cow casein allergens.

Milk Proteins	Identified Allergic Epitope Peptide Fractions
Casein proteins (Bos d 8)	The whole casein fraction
αS1-casein (Bos d 9)	16–35, 17–36, 19–30, 39–48, 69–78, 93–98, 93–102, 109–120, 123–132, 139–154, 141–150, 159–174, and 173–194. The whole molecule and the larger fractions have the highest effect
αS2-casein (Bos d 10)	31–44, 43–56, 83–100, 93–108, 105–114, 117–128, 143–158, 157–172, 165–188, and 191–200
β-casein (Bos d 11)	1–16, 45–54, 55–70, 83–92, 107–120, 135–144, 149–164, 167–184, and 185–208
κ-casein (Bos d 12)	15–24, 37–46, 55–80, 83–92, and 105–116

Data source [191].

## Data Availability

All data generated or analyzed during this study are included in this published article.

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
