# Peer review of "Caseins: Versatility of Their Micellar Organization in Relation to the Functional and Nutritional Properties of Milk"

_molecules, 2023, doi:10.3390/molecules28052023_

Round 1

Reviewer 1 Report

This manuscript could be considered for publication after conducting a major revision. I am hereby returning the paper to you, pointing out comments to the authors:

Keywords:

Remove and

As the authors have a comprehensive review of milk caseins, the reviewer is excepted to see milk type (A and B) and other milk functions, for example, ethanol stability, heat sensitivity for camel milk, and other mammalian milk in the revised manuscript.

Table 6 could be updated with other identified bioactive peptides other than bovine milk.

Fig 2, Fig 6, etc have methodological parts, It’s better to have this part organized in the manuscript.

Author Response

  1. As the authors have a comprehensive review of milk caseins, the reviewer is expected to see milk type (A1 and A2) and other milk functions, for example, ethanol stability, heat sensitivity for camel milk, and other mammalian milk in the revised manuscript.

Thank you for your comments, some information on milk types has been added to lines 510-517

 Information on ethanol stability and heat sensitivity is added to lines 591-607

  1. Table 6 could be updated with other identified bioactive peptides other than bovine milk.

Thank you for your comment, some identified bioactive peptides have been added to Table 6

  1. Fig 2, Fig 6, etc have methodological parts, it’s better to have this part organized in the manuscript.

 I appreciate your feedback. We believe that when discussed underneath them, the methodological components of an image or a figure would be simpler to understand.

Reviewer 2 Report

General remarks

The manuscript’s topic is very valuable for the scientific community, but I did find some minor problems.

Additional remarks:

1. Introduction: why the African elephant is interesting? Why had chosen this animal? Please explain it!

Table 1: better if apply more adequate references in this Table!

Lines 80-84: this paragraph only true to cow and camel milk! Please clarify it!

Line 119: not all milk components are secreting in the mammalian gland!

3.1: subtitle incl. isoelectronic point but not discussed in this subsection, but find only in 3.2 subsection!

line 343: the given range of cow milk pH value is a little bit higher than the normal pH values (6.5-6.7)!

lines 350-354: please add a reference!

Author Response

  1. Introduction: why the African elephant is interesting? Why had chosen this animal? Please explain it.

The African elephant is interesting because it is the only mammal in the study that lacks both (αS1-, αS2-) casein making them uniquely different. We choose animals with extreme casein variations.

2. Table 1: better if apply more adequate references in this Table!

 Thank you. more references have been added in Table 1.

3. Lines 80-84: this paragraph is only true to cow and camel milk! Please clarify it!

The statement from this paragraph is true for cows and camels, but for African elephants, even though dairy products from African Elephants are rare to come by.

4. Line 119: not all milk components are secreting in the mammalian gland!

Thank you for your comment, the statement has been rewritten for clarity (lines 123-126)

4. subtitle incl. isoelectronic point but not discussed in this subsection but find only in the 3.2 subsections!

Thank you for your observation, the subtitle 3.1 has been altered. Thus, isoelectronic has been moved to subsection 3.2 line 238

5. line 343: the given range of cow milk pH value is a little bit higher than the normal pH values (6.5-6.7)!

Thank you for your observation cow milk pH value range has been changed (line 344)

6. lines 350-354: please add a reference!

Thank you for your comment, a reference has been added to line 355

Reviewer 3 Report

This work illustrated the genetic and biosynthesis, molecular structure, composition and structure, functional and nutritional properties of milk caseins from four species, namely cow, camel, human and African elephant. In addition, the authors want to figure out the influence of the genetic and structure information on functional and nutritional properties of milk caseins, which was beneficial to the development of novel milk-based food products. However, this study has several problems as following:

 1. There were a few integrated and original figures and tables since most of the figures and tables were from other literatures, which implied shortage of independent and critical thinking of the review.  

2. The functional and nutritional properties of milk caseins more focused on caseins from cow and camel, less on caseins from human and African elephant, inconsistent with the four species in previous genetic and structure parts, which was not beneficial for investigating the mechanism of structure affecting functional and nutritional properties of caseins.

3. The authors clarified no clear view or statements in the review, especially in the conclusion, thus it probably did not brought new information for readers about casein micellar organization in relation to the functional and nutritional properties of milk as stated in the title. The mechanism of structure affecting functional and nutritional properties of caseins should be explored further based on available information.

4. Line 55: The authors compared the structures of the casein protein sequences in four selected animal species, namely, cow, camel, human, and African elephant, but why didn't discuss the nutritional function of elephant casein in the manuscript?

5. Line 56: “indicating that the casein micelles have highly flexible structures that do not require the presence of all four caseins”? needs to be supported by more references.

6. Line 691Does casein polymorphism affect milk's nutritional properties only in diabetes, autism, and allergy? And the authors didn't mention theapplication in the manuscript, but the title is Nutritional properties and applications.

7. Line 834: In the conclusion section, the authors didn't conclude a constructive or innovative conclusion.

Author Response

1. There were a few integrated and original figures and tables since most of the figures and tables were from other literature, which implied a shortage of independent and critical thinking in the review

Thank you! your comment is noted.

2. The functional and nutritional properties of milk caseins are more focused on caseins from cows and camel, and less on caseins from humans and African elephants, inconsistent with the four species in previous genetic and structure parts, which was not beneficial for investigating the mechanism of a structure affecting functional and nutritional properties of caseins.

 Thank you, your comment is noted

3. The authors clarified no clear view or statements in the review, especially in the conclusion, thus it probably did not bring new information for readers about casein micellar organization in relation to the functional and nutritional properties of milk as stated in the title. The mechanism of a structure affecting the functional and nutritional properties of caseins should be explored further based on available information.

Thank you, your comment is noted.

4. Line 55: The authors compared the structures of the casein protein sequences in four selected animal species, namely, cow, camel, human, and African elephant, but why didn't discuss the nutritional function of elephant casein in the manuscript?

Thank you for your comment, a discussion on African elephant milk has been added to lines 58-63

5. Line 56: “indicating that the casein micelles have highly flexible structures that do not require the presence of all four caseins”? needs to be supported by more references.

Thank you for your observation, the has been rewritten for clarity and some reference are added to support it line 60 & 62 

6. Line 691:Does casein polymorphism affect milk's nutritional properties only in diabetes, autism, and allergy? And the authors didn't mention the “application” in the manuscript, but the title is Nutritional properties and applications.

Thank you for your comments. Casein polymorphism is associated with many nutrition-related diseases, But in this review, our focus was limited to, Diabetes, Allergy, and Autism. more information is added on lines 720-727

7. Line 834: In the conclusion section, the authors didn't conclude a constructive or innovative conclusion.

Thank you for your comment, few lines are added in the conclusion to improve it lines 889-891.

Reviewer 4 Report

Review 10.01.2023

Caseins and versatility of their micellar organization in relation 2 to the functional and nutritional properties of milk

Ashish Runthalaa,*, Mustapha Mbyeb, Mutamed Ayyashb, Yajun Xuc and Afaf Kamal-Eldin

Reviewer comments 

Well written abstract

Line 79 the literature

Clearly presented introduction

Genetics introduction is very easy to follow in section 2

Line 118 the blood supply

Line 166 don’t start the sentence/ paragraph with a figure number

Line 191 – reference data from Figure 2.

Line 202 – reference the source of information in table 2.

Line 204 CLUSTAL W

Line 205 αs1 not alpha…

Line 231 don’t start the sentence/ paragraph with a figure number

Line 282 oC remove underline

Line 295  van der Waals

Table 3 - Instability index lower than 40 affirms the stability of a protein [177]- none of the samples are less than 40? Can you clearly refer to this in the paragraph above?

Line 328 Figure 5 Labelling has moved on figure. Why are there two cow images and 1 for human and camel?

Line 341 Ginger & Grigor, 1999 – this is a very old reference, is there nothing more up to date?

Line 355 don’t start the sentence/ paragraph with a figure number

Line 358 full stop

Line 385 According to (mention Authors) [62]

Line 398 don’t start the sentence/ paragraph with a figure number

Line 414 According to (mention Authors) [69]

Line 419 models by (mention Authors) [67]

Line 424 (mention Authors) [71] suggested that the casein  

Line 443 (see below) remove this and refer to figure or table.

Line 482 Thus, the previous suggestion by (mention Authors) [76]

Line 497 compared to the size of casein micelles

Line 522 don’t start the sentence/ paragraph with a figure number

Line 534 – have you permission to use these images?

Line 537 (Dallas et al., 2014) why is this format given and I could not find Dallas et al 2014 in list of references?

Line 540 full stop

Line 595 residues

Line 618 don’t start the sentence/ paragraph with a figure number

Line 631 reference statement

Line 637 reference table content source (actually the reference has moved to line 639).

Line 649 full stop

Line 655 have you permission to use the table content or did you adapt from original source?

Line 677 full stop

Line 690 full stop

Line 713 Figure different font colour

186 references, well researched review paper.

Final comments

This review paper is well written, structurally easy to follow and good use of tables and figures throughout, just please make sure they are all referenced.

The contrast between the casein of bovine, human, African elephant and camel is an interesting one. I would have liked to hear a simple statement on the volume of milk each mammal can produce per annum. Also, which of the milks mentioned are being used to produce cheese and youghurt.

Cows, sheep and goats exist internationally in many countries, whereas camels and elephants do not. What are the implications there?

The links to camel milk as an antidiabetic treatment is an unusual one and also the link of bovine milk to Autism is something I am not familiar with.

The review paper is an interesting read, condenses useful information on physicochemical properties and biological characteristics of caseins from multiple sources.

A short list of abbreviations would be useful i.e. GMP, CCP, PKC, DPP, BCM, CMA etc…

Author Response

1. Line 79 the literature.

Thank you for your observation, the has been added line 84

2. Clearly presented introduction

Thank you!The 

3 Genetics introduction is very easy to follow in section 2

Thank you

4. Line 118 the blood supply

Thank you, the statement has been, rewritten as requested by reviewer 2 Lines 123-126

5. Line 166 don’t start the sentence/ paragraph with a figure number

Thank you, the sentence has been, rewritten in Lines 172-174

6. Line 191 – reference data from Figure 2.

Thank you, we have cited the reference. Line 207

7. Line 202 – reference the source of information in table 2.

Thank you, we have cited the reference line 213

8. Line 204 CLUSTAL W

Thank you, it is corrected line 211

9. Line 205 αs2 not alpha…

Thank you, the word has been corrected in line 212

10. Line 231 don’t start the sentence/ paragraph with a figure number

Thank you the sentence has been rewritten lines 240-241

11. Line 282 oC remove the underline

Thank you, the underline has been, removed from lines 291

12 Line 295 van der Waals

Thank you, the Waals have been, corrected in line 304

13. Table 3 - Instability index lower than 40 affirms the stability of a protein [177]- none of the samples are less than 40? Can you clearly refer to this in the paragraph above?

Thank you for your observation, the statement has been added in paragraph lines 305-307

14 Line 328 Figure 5 Labelling has moved on the figure. Why are there two cow images and 1 for a human and a camel?

Thank you for your observation the labeling has been moved back to its rightful position, and the images have been corrected. Figure 5

15. Line 341 Ginger & Grigor, 1999 – this is a very old reference, is there nothing more up to date?

Thank you for the observation the reference has been updated to lines 342

16 Line 355 don’t start the sentence/ paragraph with a figure number

Thank you, the sentence has been rewritten lines 356-357

17 Line 358 full stop

Thank you for your observation. A Full stop has been added to line 362

18. Line 385 According to (mention Authors) [62

Thanks for your observation the Authors have been added. Line 387

19 Line 398 don’t start the sentence/ paragraph with a figure number

Thank you, the sentence has been rewritten lines 399-400

21. Line 414 According to (mention Authors) [69]

Thank you for your observation the Author has been added. Line 415

22 Line 419 models by (mention Authors) [67]

Thank you, the author has been added. Line 420

23 Line 424 (mention Authors) [71] suggested that the casein  

Thank you, the author has been added line 425

24. Line 443 (see below) remove this and refer to the figure or table.

Thanks for your observation the sentence has been rewritten lines 443-445

25. Line 482 Thus, the previous suggestion by (mention Authors) [76]

Thank you, the author has been added lines 484

26 Line 497 compared to the size of casein micelles

Thank you for the observation to has been added line 498

27. Line 522 don’t start the sentence/ paragraph with a figure number

Thank you for your comment, the sentence has been rewritten to lines 532-533

28. Line 534 – have your permission to use these images?

Thank you. Yes, the author has permission to use the image in figure 8. 

29. Line 537 (Dallas et al., 2014) why is this format given and I could not find Dallas et al 2014 in list of references?

Thank you for your observation, the correct format has been used and the reference has been added to the reference list. Line 547

30. Line 540 full stop

Thank you, a full stop has been added to line 552

31. Line 595 residues

Thank you, residues have been added to line 623

 32 Line 618 don’t start the sentence/ paragraph with a figure number

Thank you, the sentence has been rewritten lines 646-647

33. Line 631 reference statement

Thank you, some reference has been, added to support the statement line 660

34. Line 637 reference table content source (actually the reference has moved to line 639).

Thank you for your observation the reference has been moved to Line 667

35. Line 649 full stop

Thank you, a full stop has been, added to line 677

36. Line 655 have your permission to use the table content or did you adapt from original source?

Thank you, the authors have permission to use table 6.

37. Line 677 full stop

Thank you, a full stop has been, added. line 705

38. Line 690 full stop

Thank you, a full stop has been, added. line 718

39. Line 713 Figure different font color

Thank you for observing this from the original figure.

40. 186 references, well-researched review paper.

Thank you for your comments.

41. Final comments

42. This review paper is well written, structurally easy to follow, and good use of tables and figures throughout, just please make sure they are all referenced.

Thank you for your kind words. 

43. The contrast between the casein of bovine, human, African elephant, and camel is an interesting one. I would have liked to hear a simple statement on the volume of milk each mammal can produce per annum. Also, which of the milks mentioned are being used to produce cheese and yogurt.

Thank you for your kind words. The review focused more on the casein versatility of the various animals studied.

Cheese and yogurt can be made from any mammal's milk, including elephant and human milk. Commercially produced cheese and yogurt are primarily made from milk from cows, goats, and sheep. The production of dairy products from horse milk and camel milk is also emerging.

44. Cows, sheep, and goats exist internationally in many countries, whereas camels and elephants do not. What are the implications there?

The study focused on the various caseins components differences in studied animals (Cows, sheep, and goats) that have closely related milk casein. Thus, a study on cow caseins could have the same implications on sheep and goats.

45. The link to camel milk as an antidiabetic treatment is an unusual one and the link of bovine milk to Autism is something I am not familiar with.

There are various studies both in animals and humans that showed the potential of camel milk to have anti-diabetic effects on both Type 1 and Type 2 diabetic patients.  You may be interested check out the following articles (Abdulrahman et al. 2016; Agrawal et al. 2011; Agrawal et al. 2007; Al-Numair, et al., 2011; Mohamad et al. 2009) just to name a few.

47. The review paper is an interesting read and condenses useful information on physicochemical properties and biological characteristics of caseins from multiple sources.

Thank you very much for the detailed review

48. A short list of abbreviations would be useful i.e. GMP, CCP, PKC, DPP, BCM, CMA etc…

Thank you for your valuable comment but since all the abbreviations have been spelled out in full when they were used for the first time, we don’t think it is necessary to list them. 

Round 2

Reviewer 1 Report

The authors have done a lot of work on the paper and made substantial improvements. However, this manuscript could be accepted for publication after minor revision. I am hereby returning the paper to you, pointing out comments to the authors:

1/ The authors need to pay attention to the reviewer's second comment related to Table 6.

2/ Authors are required to check recently added references, especially No 129, 132, 133, etc.

Author Response

1. Table 6 could be updated with other identified bioactive peptides other than bovine milk.

We appreciate your comments. Some bioactive peptides identified from camels have been added to the table in red, and those from humans have been added in blue. Unfortunately, despite searching for information on the bioactive peptides of African elephants, the authors were unable to find any.

2/ Authors are required to check recently added references, especially No 129, 132, 133, etc.

Thanks for your valuable comments. The references have been rechecked.

Reviewer 3 Report

I have read over your manuscript and found it has been well revised and greatly improved. However, the current manuscript cannot meet Journal's requirements for its high standards. I hope the author will continue to enrich and improve the research content for breakthroughs and innovations.

Author Response

I have read over your manuscript and found it has been well-revised and greatly improved. However, the current manuscript cannot meet Journal's requirements for its high standards. I hope the author will continue to enrich and improve the research content for breakthroughs and innovations.

Thanks for your valuable comments. The authors appreciate your service and will continue to work hard to achieve breakthroughs.